# Learning Chaotic Dynamics with Embedded Dissipativity

## Abstract

Chaotic dynamics, commonly seen in weather systems and fluid turbulence, are characterized by their sensitivity to initial conditions, which makes accurate prediction challenging. Despite its sensitivity to initial perturbations, many chaotic systems observe dissipative behaviors and ergodicity. Therefore, recently various approaches have been proposed to develop data-driven models preserving invariant statistics over long horizons. Although these methods have shown empirical success in reducing instances of unbounded trajectory generation, many of the models are still prone to generating unbounded trajectories, leading to invalid statistics evaluation. In this paper, we propose a novel neural network architecture that simultaneously learns a dissipative dynamics emulator that guarantees to generate bounded trajectories and an energy-like function that governs the dissipative behavior. More specifically, by leveraging control-theoretic ideas, we derive algebraic conditions based on the learned energy-like function that ensure asymptotic convergence to an invariant level set. Using these algebraic conditions, our proposed model enforces dissipativity through a ReLU projection layer, which provides formal trajectory boundedness guarantees. Furthermore, the invariant level set provides an outer estimate for the strange attractor, which is known to be very difficult to characterize due to its complex geometry. We demonstrate the capability of our model in producing bounded long-horizon trajectory forecasts that preserve invariant statistics and characterizing the attractor, for chaotic dynamical systems including Lorenz 96 and a truncated Kuramoto–Sivashinsky equation.

## 1 Introduction

Chaos, characterized by exponential divergence after infinitesimal initial perturbations, is ubiquitous in a variety of complex dynamical systems including climate models (Lorenz, 1963) and turbulence in fluids (Kuramoto, 1978; Ashinsky, 1988). The exponential separation makes it challenging to accurately predict their trajectories using data-driven methods. Despite intrinsic instability, many chaotic systems are dissipative, i.e., their trajectories will converge to a bounded positively invariant set (Stuart & Humphries, 1998), which is also known as a strange attractor. Additionally, trajectories of dissipative chaos will visit almost every state on the attractor, resulting in ergodicity and invariant statistics (Guckenheimer & Holmes, 2013). As a result, recent progress in dissipative chaotic systems learning has been focused on applying deep learning to learn a dynamics emulator that preserves invariant statistics over a long forecast horizon instead of pointwise accurate prediction.

In order to obtain meaningful statistics from the trajectories generated by the learned dynamics emulator, it is often required to roll out the model prediction for a long time horizon. Similar to the true system, an accurate model aiming at preserving the invariant statistics needs to visit every possible state on the strange attractor. Since the training dataset is limited, the model needs to generalize well to states not seen during training. More specifically, it is crucial that the learned model always generate bounded trajectories from all initial states, otherwise the statistics evaluation will be invalid. However, it can be challenging to keep predicted trajectories bounded for a sufficiently long time, as these models are trained to match the chaotic behaviors observed in the training data.

Despite empirical success in developing data-driven models that preserve invariant statistics well, many of these models still suffer from the risk of generating unbounded trajectories. For specific regression models such as multi-level ad hoc quadratic regression models, theoretical limitations have

been investigated by Majda & Yuan (2012), where they found these models could lead to finite-time blowup or unstable statistical solutions, rendering the prediction unreliable. For sequential time-series models, it was shown by Mikhaeil et al. (2022) that training a recurrent neural network (RNN) for chaotic system prediction can lead to unbounded gradients. Other than RNN, reservoir computing (RC) has been a popular choice in modeling chaotic systems due to its relatively simple structure and computation efficiency, which helps avoid unbounded gradient issues occured in RNN-based approaches (Wikner et al., 2022). Although RC models have shown good prediction performance in capturing invariant statistics, Lu et al. (2018); Pathak et al. (2017) have observed that they are prone to step on a state not seen during training and experiencing finite-time blowup, which reduces the length of reliable forecast.

**Related Work.** To gain deeper understanding of this issue and encourage boundedness in trajectories over long forecast horizons, various approaches have been proposed under different settings. For RC-based models, Shirin et al. (2019) proposes a stability analysis approach based on Lyapunov methods to estimate the region of attraction of a fixed point, as an effort to quantify the globally stable region of the learned model. To improve RC-based model's stability over long time horizons, a noise-inspired regularization has shown empirical success in promoting stability (Wikner et al., 2022), and it is also shown by Haluszczynski et al. (2020) that the predicted trajectory stability can be improved empirically by reducing network size and removing certain nodes.

In addition to modifying an architecture or adding noise in the training data, recent literature has shown that empirical improvement in stability can also be achieved by prioritizing on learning statistics over pointwise prediction. Platt et al. (2023) propose an RC-based method that constrains the model to match the Lyapunov exponent spectrum and fractal dimension of the true system. Based on the neural operator framework, Jiang et al. (2024) propose to add statistics-based regularization to the loss function through either expert knowledge or contrastive learning, both of which have shown to improve the model's prediction for the invariant statistics. Despite empirical evidence that these methods can reduce the instances of unbounded trajectory generation, there is still a risk of generating unbounded trajectory without any formal treatment regarding dissipativity, which prevents valid statistics evaluation.

Instead of empirical improvement in reducing unbounded trajectory generation instances, a few works in the literature have been developed to ensure trajectory boundedness by enforcing dissipativity in the learned model. Based on quadratic nonlinear models, Majda & Harlim (2012) propose a physics-constrained multi-level regression method which employs a linear system and nonlinear system component, and ensures the model to be dissipative by careful construction of the nonlinear term. Based on the neural operator framework, Li et al. (2022) propose to enforce dissipativity in the model by explicitly altering the flow of the learned dynamics emulator in a pre-specified region of the state space. Although both methods show success in predicting chaotic systems with boundedness guarantees, they often require expert knowledge in constructing the model to pre-specify the dissipativity condition, which can be challenging to obtain for complex systems.

**Our Key Contributions.** In this paper, we propose an approach to learn a neural network model for dissipative chaotic systems that is guaranteed to generate bounded trajectories over long forecast horizons. Unlike previous approaches that require prior information about the system, we simultaneously learn a dynamics emulator that is formally guaranteed to be dissipative and an energy function that governs the dissipative behavior. Our key contributions are three-fold:

1. By leveraging Lyapunov methods in control theory, we introduce level sets of Lyapunov functions as an attractor that the dissipative system trajectory converges to, which allows us to derive algebraic conditions for dissipativity that are computationally efficient to verify (Section 3).

2. We enforce the aforementioned algebraic condition in the neural network model, by constructing a projection layer that ensures the learned dynamics emulator to be dissipative (Section 4.1).

3. By incorporating regularization loss of the invariant level set volume in the training process, our model also learns an outer estimate of the strange attractor which is very difficult to characterize due to its complex geometry (Section 4.2).

Additionally, we first illustrate the novel stability projection layer that ensures trajectory boundedness through a numerical experiment on the Lorenz 63 system. We demonstrate the effectiveness

of our proposed model in generating bounded trajectories that preserve invariant statistics over long forecast horizons, and in learning an outer estimate of the strange attractor, on the Lorenz 96 system and a truncated Kuramoto–Sivashinsky equation.

## 2 BACKGROUND AND PROBLEM FORMULATION

Consider a chaotic dynamical system described as a finite-dimensional ODE,

$$\dot{x}(t) = f(x(t)), \tag{1}$$

where $x(t) \in \mathbb{R}^n$ is a $n$-dimensional vector which represents the state of the dynamical system at time $t$, and $f : \mathbb{R}^n \to \mathbb{R}^n$ is a nonlinear function that governs the dynamics. The objective is to construct a neural network dynamics emulator $\hat{f} : \mathbb{R}^n \to \mathbb{R}^n$, such that by solving the initial value problem

$$\dot{\hat{x}}(t) = \hat{f}(\hat{x}(t)), \quad \hat{x}(0) = x(0) \tag{2}$$

from a given initial condition $x(0)$, the solution $\hat{x}(t), t \in [0, T]$ approximates the true solution $x(t)$.

**Characteristics of Chaotic Dynamical Systems.** As discussed in (Strogatz, 2018), a chaotic system typically exhibits the following three properties, albeit there is no universally accepted definition:

1. "Aperiodic long-term behavior": There is a set with non-zero measure of initial conditions from which the system's trajectory does not converge to a fixed point or periodic orbits or quasiperiodic orbits.

2. "Deterministic": There is no stochasticity in either the input or parameters of the system.

3. "Sensitive dependence on initial conditions": Small perturbations in the initial conditions may lead to exponential separation in the trajectory in certain regions of the state space.

Among the three properties listed above, the third one is the most quantifiable and also well known as the "butterfly effect" (Lorenz, 1963). More formally, the "sensitive dependence on initial conditions" is quantified as the Lyapunov exponent being positive. In what follows, we give an informal definition for the Lyapunov exponent and explain its implications for prediction problems.

***Intuitive Definition for Lyapunov Exponent*** *(Vulpiani, 2010): Suppose there are two trajectories of the system whose initial conditions differ by $\delta x_0$. The Lyapunov exponent $\lambda$ describes the rate of exponential growth of the distance between the two trajectories, i.e., $\|\delta x(t)\| \approx exp(\lambda t)\|\delta x_0\|, \forall t \in [0, t_0]$ for some finite time $t_0 > 0$.*

As a signature of chaotic systems, the Lyapunov exponent being positive implies that neighboring trajectories diverge exponentially. Note that in the presence of a positive Lyapunov exponent, if the initial condition is measured with a tiny error $\|\delta_0\|$, the prediction error will grow above a given small threshold $\epsilon$ in finite time of $O(\ln(\epsilon/\|\delta_0\|)/\lambda)$. In other words, in regions where exponential separation from neighboring trajectories occur, accurate prediction of chaotic systems is almost impossible in the long run (Strogatz, 2018).

Despite the fact that neighboring trajectories may diverge exponentially in certain regions of the state space, it is important to make the distinction between chaos and instability. As an example in (Strogatz, 2018), the scalar system $\dot{x} = x$ is deterministic and exhibits sensitive dependence on initial conditions. However, this is not a chaotic system, as every trajectory grows to infinity and never returns, which makes infinity an attracting fixed point. Such global unstable behaviors exclude the system from being chaotic, because it is not aperiodic in the long term.

**The Strange Attractor and Invariant Statistics.** As discussed earlier, global instability often disqualifies a system from being chaotic. One natural question is what it means to have exponential separation of neighboring trajectories while not being globally unstable. An intuitive explanation is that the exponential divergence only occurs locally within certain regions of the state space, which corresponds to the "strange attractor". Due to the disagreement in the literature on a precise definition of the strange attractor, we list a few important characteristics of the strange attractor instead of giving a formal definition (Strogatz, 2018). Interested readers may refer to (Milnor, 1985; Guckenheimer & Holmes, 2013) for detailed discussions on the subtleties of such definitions.

The strange attractor $M$ for a chaotic system has the following characteristics (Strogatz, 2018):

1. $M$ is a positively invariant set that attracts trajectories that start in an open neighborhood.

2. $M$ minimal, i.e., there is no proper subset of $M$ that satisfies the above property.

3. Exponential separation of neighboring trajectories occurs with all initial conditions in $M$.

The attractivity and invariant set properties of the strange attractor make sure trajectories that are sufficiently close to the attractor will converge to it asymptotically. Intuitively, this behavior prevents exponential separation from growing to infinity by attracting trajectories to a local region.

Furthermore, the third property requires all neighboring trajectories starting on the strange attractor to separate exponentially. More importantly, it leads to the fact that the system will visit almost every state on the attractor, in other words, the system will be ergodic once it enters $M$. As a result, the strange attractor provides invariant statistics of the system (Guckenheimer & Holmes, 2013). Given the challenges in predicting such systems rooted in their chaotic nature, the existence of the strange attractor allows for the possibility of evaluating the statistical properties of the system, rather than only making pointwise predictions.

**Dissipative Chaotic Systems.** So far we have discussed the importance of the strange attractor in preserving invariant statistics, but we have not addressed the existence and boundedness of the strange attractor, which are crucial premises for evaluating invariant statistics of the chaotic system. In general, not every chaotic system has a strange attractor, especially if they are Hamiltonian systems (Ott, 2002). However, over the course of study on chaos, a large class of chaotic systems, dissipative chaos, have been known to have strange attractors, which are bounded and positively invariant sets that attract trajectories in an open neighborhood. As a result, we will focus on this class of chaotic systems in this paper.

In fact, many chaotic systems are dissipative, including the well-studied Lorenz 63 system (Lorenz, 1963) and the Kuramoto–Sivashinsky equation (Kuramoto, 1978). Note that dissipativity is a property of dynamical systems that describes its evolution over time, independent of whether the system is chaotic or not. Here we provide a formal definition of dissipativity and then discuss its implications for learning chaos.

**Definition 1.** *We say that the system (1) is **dissipative** if there exists a bounded and positively invariant set $M \subset \mathbb{R}^n$ such that $\lim_{t \to \infty} dist(x(t), M) = 0$, where $dist(x(t), M) = \inf_{y \in M} \|x(t) - y\|$. In other words, every trajectory of the system will converge to $M$ asymptotically, and stays within $M$ once it enters. $M$ is said to be **globally asymptotically stable**.*

The definition is derived based on the one introduced by Stuart & Humphries (1998), where they stated dissipativity as every trajectory will enter $M$ eventually. Here, we quantify this behavior using the notion of asymptotic stability and trajectory distance to $M$. As alluded to earlier, a chaotic system being dissipative ensures that the trajectory always stays bounded and converges to its strange attractor, in which every state is visited almost uniformly, consequently producing invariant statistics for evaluation.

**Our Objective: Learning Dissipative Chaotic Dynamics.**

As discussed extensively in this section, we choose to focus on developing models for dissipative chaotic dynamics because of their invariant statistics. As seen in the recent literature, a tractable goal is to learn a dynamics emulator producing trajectories that match the invariant statistics observed in the original system over long horizons. Although matching statistics is slightly less challenging than pointwise prediction, the chaotic nature of the system, namely exponential separation on the strange attractor, still poses challenges for machine learning models.

More specifically, since the system trajectory will traverse through every state on the attractor infinitely often, training directly from trajectory data will often lead to an unstable learned dynamics model as it is trying to match the chaotic behaviors observed in the data. Since the training dataset is limited, during testing the model might not generalize well on the states inside the strange attractor that are not seen during training, resulting in finite-time blowup as reported by Lu et al. (2018). While as surveyed in the introduction, numerous methods have been proposed to address this issue and achieved empirical success in reducing the instances of unbounded trajectory generation, there is still a risk of generating unbounded trajectory without any formal treatment regarding dissipativity, which prevents valid statistics evaluation.

In this paper, we aim to develop neural network-based models with embedded dissipativity. By establishing the connection between dissipativity and energy and leveraging energy-like Lyapunov functions in control theory, we propose a model architecture that is designed to be dissipative, and therefore always generates bounded solutions for the initial value problem (2).

## 3  ALGEBRAIC DISSIPATIVITY CONDITIONS AND ATTRACTOR ESTIMATION

In Section 2, we have discussed in detail why learning dissipative chaotic systems is a meaningful and tractable problem, and we identified the challenges involved. More specifically, dissipativity in chaos ensures the existence of a strange attractor leading to invariant statistics, but in order to obtain valid statistics, the learned models need to generate bounded trajectories over long time horizons. To develop models with embedded dissipativity, we first need to understand theoretical conditions that make a dynamical system dissipative. Note that dissipativity describes a system's energy behavior over time, independent of whether the system is chaotic or not. In this section, we focus on deriving algebraic conditions that are computationally efficient through the connection between dissipativity and energy, which are crucial for our proposed architecture that guarantees dissipativity.

### 3.1  ENERGY-BASED CONDITIONS FOR DISSIPATIVITY

Dissipativity, as the name suggests, has a close relationship with energy in a dynamical system. Intuitively, a dissipative system will lose energy over time, which corresponds to the trajectory converging to a bounded set. Although the trajectory convergence behavior is quantitatively stated in Definition 1, given a system $\dot{x} = f(x)$ known to be dissipative with access to the true dynamics $f$, it is still challenging to quantify the set $M$ that the trajectory converges to. In the specific context of dissipative chaos, there has been a body of literature that tries to address this issue by studying invariant manifolds, volume contraction, and attempting to characterize the strange attractor (Stuart & Humphries, 1998). Despite the rigorous treatment and the progress over the years that help us understand the strange attractor, these characterizations are often stated in abstract mathematical concepts and a descriptive manner that is intractable to computationally verify, e.g., (Milnor, 1985). Given our goal of enforcing dissipativity in neural network models, it is crucial to first derive computationally efficient conditions that ensure a system is dissipative.

In control theory, the concept of Lyapunov functions has been used extensively to formalize asymptotic stability of dynamical systems, which are also known as "energy-like" functions due to strong connections with the mechanical energy of the system. More importantly, by leveraging the level set of such functions, numerous computationally tractable conditions have been derived and extensively used in designing practical controllers to ensure a system's asymptotic stability to equilibrium points (Khalil, 2002). By generalizing asymptotic stability with respect to an equilibrium point to a level set of a Lyapunov function, we derive computationally efficient conditions that ensure dissipativity in a dynamical system.

Recall in Definition 1, a dissipative system requires the existence of a bounded set $M$, which satisfies (1) $M$ is an invariant set (2) the system is globally asymptotically stable towards $M$. By reducing the definition to the existence of a Lyapunov function $V$ and choosing $M$ to be a level set of $V$, i.e., $M(c) = \{x : V(x) \leq c\}$ where $c > 0$ corresponds to the energy level, we derive the following conditions for invariance and asymptotic stability of $M(c)$ in Proposition 1 and 2, respectively.

**Proposition 1** (invariant level set)**.** *For a dynamical system in (1), suppose there is a continuously differentiable scalar-valued function $V : \mathbb{R}^n \to \mathbb{R}$ and a constant $c > 0$, such that*

$$\forall x \in \{x \in \mathbb{R}^n : V(x) > c\}, \dot{V}(x) = \frac{\partial V}{\partial x} f(x) \leq 0.^1$$

*Then the level set $M(c) = \{x : V(x) \leq c\}$ is a positively invariant set for the system (1).*

**Proposition 2** (asymptotic stability)**.** *For a dynamical system in (1), suppose there is a lower-bounded continuously differentiable scalar-valued function $V : \mathbb{R}^n \to \mathbb{R}$ and a constant $c > 0$, such that*

*(1) $\forall x \in \{x \in \mathbb{R}^n : V(x) > c\}, \dot{V}(x) < 0$; (2) $V$ is radially unbounded.*

*Then the level set $M(c) = \{x : V(x) \leq c\}$ is globally asymptotically stable.*

---

[1] Here $\frac{\partial V}{\partial x}$ refers to the row vector $[\frac{\partial V}{\partial x_1}, ..., \frac{\partial V}{\partial x_n}]$.

The proofs for both propositions are included in Appendix A. We include the following illustrative figures to sketch the intuitions behind the conditions in Proposition 1 and 2.

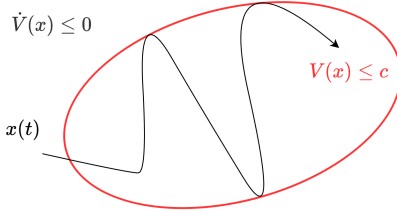 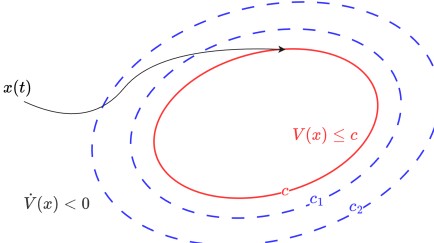

(a) The level set boundary serves as a barrier since the trajectory cannot gain energy outside. Once entering $M(c)$, the trajectory will be confined within.

(b) The trajectory loses energy over time outside $M(c)$ because $\dot{V}(x) \leq 0$, resulting in convergence to the level set $M(c)$ eventually ($c_2 > c_1 > c > 0$).

Figure 1: Illustrations for positive invariance and global asymptotic stability conditions.

## 3.2 An Algebraic Condition for Dissipativity and Attractor Outer Estimation

Compared to Definition 1, the conditions derived in the propositions above are much more quantitative. However, verifying these conditions for a given system in (1) computationally is not trivial; for example, the negative semi-definite condition for $\dot{V}(x)$ is defined only on certain part of the state space outside the level set $M(c)$ in Proposition 1. Inspired by s-procedure in sum-of-squares programming (Blekherman et al., 2012), we can replace these conditions with a single algebraic condition that is independent of the state, which is more computationally tractable.

**Theorem 1.** *Suppose there exists a lower-bounded radially unbounded $C^1$ function $V : \mathbb{R}^n \to \mathbb{R}$ and a constant $c > 0$ such that for the dynamical system in (1),*

$$\forall x \in \mathbb{R}^n, \dot{V}(x) + V(x) - c \leq 0. \tag{3}$$

*Then the system (1) is dissipative and $M(c)$ is globally asymptotically stable.*

*Proof.* The condition (3) implies that $\forall x \in \mathbb{R}^n$ such that $V(x) > c$, $\dot{V}(x) \leq -(V(x) - c) < 0$. Therefore, by Proposition 1 and 2, the level set $M(c)$ is both globally asymptotically stable and positively invariant. $\square$

**Strange Attractor Outer Estimation.** Recall the characteristics of the strange attractor discussed in Section 2, if only the first two properties are satisfied by a set $S$, then this set is known as "the attractor" (Strogatz, 2018). The difference from the strange attractor is that the attractor does not require the exponential separation of neighboring trajectories everywhere, hence it is also a superset of the strange attractor. Note that if a set $S \subset M(c)$ is an attractor, then $M(c)$ is both invariant and globally asymptotically stable. Therefore, in addition to certifying a system to be dissipative, the level set $M(c)$ also provides an outer approximate of the original attractor, which is consequently an outer estimate of the strange attractor as well.

In practice, characterizing the strange attractor is challenging due to its complex geometry (Milnor, 1985). In addition to empirically reproducing the strange attractor using the learned model similar to (Li et al., 2022; Jiang et al., 2024), our method also provides a level set outer estimate for the strange attractor by learning the Lyapunov function $V$ and level set parameter $c$.

**Importance of the Algebraic Condition.** The condition in (3) is strictly stronger than the conditions in Proposition 1 and 2. Although the condition is slightly more conservative, the computational tractability of obtaining a state-independent algebraic condition is well worth the trade-off. More importantly, this algebraic condition is crucial for constructing our proposed model architecture that ensures the learned system is dissipative, which will be discussed in detail in the next section.

## 4 METHODOLOGY

Now we propose our architecture that simultaneously learns a dynamics emulator and a Lyapunov function $V$, with the former guaranteed to be dissipative by enforcing the condition in (3) through the construction of a projection layer. The overall structure is illustrated below:

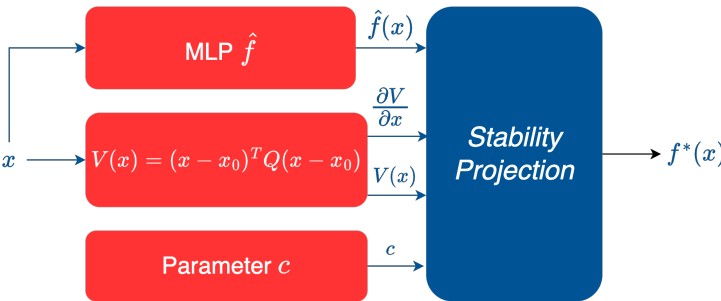

Figure 2: An illustration of the proposed neural network model.

Our proposed model consists of three learnable components illustrated in red: a Multi-Layer Perceptron (MLP) $\hat{f}$ that approximates the true nonlinear dynamics $f$ on the right-hand side of (1); a quadratic function $V(x)$ that serves as a Lyapunov function with learnable parameters $Q$ and $x_0$ that define the shape and center of the level sets, respectively; and a level set parameter $c > 0$. Using these three components and the gradient information $\partial V / \partial x$, we construct a "stability projection" layer that outputs a predicted dynamics emulator $f^*(x) \in \mathbb{R}^n$ which guarantees dissipativity of its corresponding dynamics $\dot{x} = f^*(x)$.

Note that we choose to parameterize the Lyapunov function $V$ as a quadratic function for simplicity, but the framework can accommodate any function satisfying the requirements stated in Theorem 1. Similarly, the framework can be adapted to function approximators that are more expressive than MLPs. In our experiments with a few chaotic systems, we have found MLPs to be sufficient for dynamics approximators. More specifically, $\hat{f}$ is trained to approximate $f$, which is most often a smooth function that can be well approximated by a MLP as extensively studied in the literature (Cybenko, 1989). Additionally, for finite-dimensional ODEs, choosing MLPs as function approximators aligns with recent line of work in neural operators (Li et al., 2022), since a neural operator collapses to an MLP in this setting.

### 4.1 THE STABILITY PROJECTION LAYER

Following Theorem 1, if $\dot{x} = f^*(x)$ satisfies the condition in (3), then the system is guaranteed to be dissipative and converge to the level set $M(c)$ defined by the learned Lyapunov function $V(x)$ and the learned constant $c > 0$. The stability projection layer is designed to modify the MLP approximator $\hat{f}$ such that the output $f^*$ produces a learned dynamical model $\dot{x} = f^*(x)$ that satisfies the condition in (3) therefore certified to be dissipative.

Intuitively, this condition informs a subspace for the vector field $f(x)$ in which the forward dynamics will be dissipative. The stability projection layer is designed to project any unconstrained dynamics approximator, in this case our MLP $\hat{f}(x)$, into such a subspace to ensure dissipativity.

More specifically, given an input $x \in \mathbb{R}^n$, the stability projection layer output $f^*(x)$ is chosen as the vector closest to the approximator $\hat{f}(x)$ under $l^2$ distance in the subspace of $\mathbb{R}^n$ defined by (3), i.e., $f^*(x)$ is the solution to the following optimization problem:

$$f^*(x) = \arg\min_{f(x)} \|f(x) - \hat{f}(x)\|_2^2 \quad \text{subject to} \quad \frac{\partial V}{\partial x} f(x) + V(x) - c \leq 0 \quad (4)$$

As discussed extensively in Section 3, the constraint of the optimization problem, which is adopted from the condition in (3), ensures the learned dynamics emulator $f^*(x)$ to be dissipative while being

computationally efficient. In addition to the fact that the condition is easily verifiable through basic arithmetic operations, the constraint is also linear in the optimization variable $f(x)$. Since the above optimization problem has a quadratic loss and a linear constraint, an explicit solution can be found and computed using ReLU activation, similar to the approach in Min et al. (2023):

$$f^*(x) = \hat{f}(x) - \frac{\partial V}{\partial x}^T \frac{\text{ReLU}\left(\frac{\partial V}{\partial x}\hat{f}(x) + V(x) - c\right)}{\|\frac{\partial V}{\partial x}\|^2} \quad (5)$$

With the stability projection layer implemented as in (5), we state the following corollary that formalizes the dissipativity of our proposed model architecture, which is a direct result of Theorem 1. A computational proof verifying $f^*(x)$ satisfies (3) $\forall x \in \mathbb{R}^n$ is included in Appendix A.

**Corollary 1.** *The learned dynamics model $\dot{x} = f^*(x)$ is a dissipative system with a bounded and positively invariant level set $M(c) = \{x : V(x) \leq c\}$. The set $M(c)$ is globally asymptotically stable, which implies every trajectory of the system is bounded and converges to $M(c)$ asymptotically.*

### 4.2 TRAINING LOSS WITH INVARIANT SET VOLUME REGULARIZATION

We consider a training dataset consisting of trajectory points which are evenly sampled at $h$ [sec] from a few ground truth trajectories initialized at randomly sampled initial conditions. Unlike (Li et al., 2022; Jiang et al., 2024), we do not assume the trajectories in the training set are already inside the attractor, which allows for more flexibility when learning unknown chaotic systems and the transition period before it reaches an invariant statistics is unknown. This is also beneficial for our proposed model to learn where to place the invariant set $M(c)$ and apply stability projection.

During training, we consider a multi-step setting, where we roll out the learned model for $T$ steps, each step sampled at $h$ [sec] using a numerical integration scheme. More specifically, given an initial condition chosen from the training dataset $x_0^{(i)}$, we forward simulate the learned system $\dot{\hat{x}}^{(i)} = f^*(\hat{x}^{(i)})$ with $\hat{x}^{(i)}(0) = x_0^{(i)}$ and obtain sampled states at the same sampling period $h$ [sec], $\hat{x}_k^{(i)} = \hat{x}^{(i)}(kh)$ at $k = 1, 2, ..., T$. By sampling $N$ such trajectory snapshots of length $T$ from the training dataset, we define the prediction loss as the MSE between the predicted rollout sequence $(\hat{x}_k^{(i)})_{k=1}^T$ and the ground truth sequence $(x_k^{(i)})_{k=1}^T$: Prediction Loss $= \frac{1}{NT}\sum_{i=1}^N \sum_{k=1}^T \|x_k^{(i)} - \hat{x}_k^{(i)}\|_2^2$.

In the stability projection layer, the quadratic Lyapunov function $V(x)$ and the level set parameter $c$ both need to be optimized during training. Although the prediction loss depends on $V$ and $c$ through the projection operator that produces $f^*(x)$, optimizing only the prediction loss may not be a well-defined optimization problem. More specifically, if we have found a level set $M(c_1)$ that is globally asymptotically and invariant, then any superset $M(c_2)$ for $c_2 > c_1 > 0$ is also globally asymptotically stable and invariant. Therefore, there could potentially be infinitely many solutions that lead to the small prediction loss.

To address this issue, we introduce a regularization loss that encourages the learned level set $M(c)$ to be as small as possible, which aligns with our goal of characterizing a tight outer estimate of the strange attractor. Towards this objective, we use the volume of the ellipsoid $M(c)$ as the regularization loss. Combining the prediction and regularization loss, we have the following training loss with a weight hyperparameter $\lambda > 0$ for balancing the regularization terms:

$$\text{Loss} = \frac{1}{NT}\sum_{i=1}^N \sum_{k=1}^T \|x_k^{(i)} - \hat{x}_k^{(i)}\|_2^2 + \lambda\text{Vol}(M(c)), \quad \text{Vol}(M(c)) = \frac{\pi^{n/2}}{\Gamma\left(\frac{n}{2}+1\right)}\sqrt{\frac{c^n}{\det(Q)}} \quad (6)$$

## 5 NUMERICAL EXPERIMENTS

We demonstrate the effectiveness of our proposed approach in learning models that are guaranteed to generate bounded trajectories with meaningful statistics in three experiments for Lorenz 63, Lorenz 96, and a truncated Kuramoto–Sivashinsky (KS) equation, with increasing level of complexity. In this section, we visualize our projection layer's effect using the Lorenz 63 system, and provide examples where our model generates bounded trajectories that preserve invariant statistics while an unconstrained model experiences finite-time blowup for Lorenz 96 and a truncated KS system. For

more quantitative evaluations regarding invariant statistics and more visualizations, please refer to Appendix B.

## 5.1 LORENZ 63

We first present numerical experiments for learning the Lorenz 63 system, which is proposed by Lorenz (1963) as a simplified model for atmospheric convection. The system can be described as a 3D ordinary differential equation (ODE) as follows,

$$\dot{x}_1 = \sigma(x_2 - x_1) \quad \dot{x}_2 = x_1(\rho - x_3) - x_2 \quad \dot{x}_3 = x_1 x_2 - \beta x_3$$

where $\sigma = 10.0, \rho = 8/3, \beta = 28.0$ has been known to generate chaotic behaviors. We trained our model on a trajectory dataset that consists of 10 trajectories of 200 steps with a sampling period of $h = 0.01$, under a multi-step prediction setting with $T = 5$.

We roll out the model for 50000 time steps using 4th order Runge-Kutta (RK4) numerical integration with a sampling rate of $h = 0.01$ [sec], and compare the trajectory with a ground truth trajectory simulated from the same initial condition. As shown in Figure 3a, the learned model generates a bounded trajectory that approximates the ground truth trajectory very well. Additionally, the learned invariant level set is a rather tight outer estimate of the strange attractor. In Figure 3b, it is observed that the learned dynamics constructed a flow map showing dissipativity explicitly. More specifically, around the invariant level set, the system flows only point into the level set, leading to invariance and asymptotic stability.

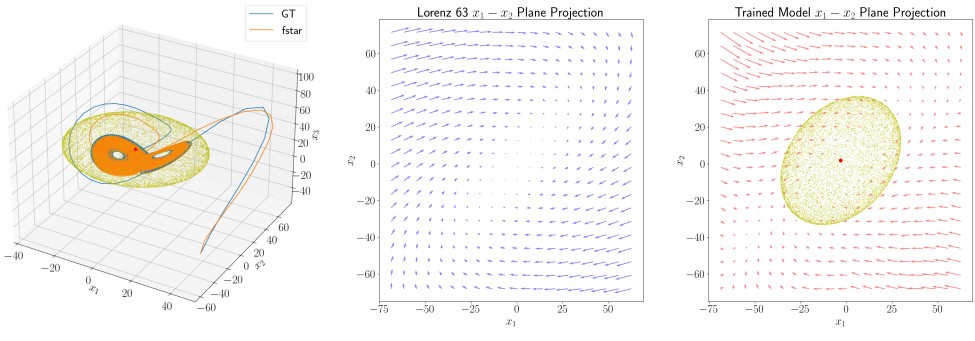

(a) 3D Trajectory rollout.          (b) Flow maps for true (left) and learned (right) dynamics.

Figure 3: Lorenz 63 (a) Trajectories generated by the learned model ("fstar") and true dynamics ("GT") are visualized along with the learned invariant set (sampled with yellow points), where the red dot is the center of the set. (b) Comparison of the flow map projected onto $x_1 - x_2$ plane.

## 5.2 LORENZ 96

We consider a slightly more complicated 5th-order Lorenz 96 system,

$$\dot{x}_i = (x_{i+1} - x_{i-2})x_{i-1} - x_i + F, \quad i = 1, 2, ..., 5, \quad x_{-1} = x_4, x_0 = x_5, x_6 = x_1$$

where $F = 8.0$ is chosen to generate chaotic behaviors. We compare a vanilla MLP model and our model based on the same MLP base architecture with components for stability projection. We train both models on a trajectory dataset consisting of 4 trajectories of 500 steps with sampling rate $h = 0.01$, under a single-step prediction setting $T = 2$.

After rolling out both the vanilla MLP and our model for 50000 steps using RK4 and sampling time $h = 0.01$ [sec], we observe that the MLP experiences finite-time blowup and generates an unbounded trajectory as shown in Figure 4a. With formal guarantees on dissipativity, our model is able to generate a bounded trajectory that approximates the true trajectory well, and learns an invariant set that provides an outer estimate for the strange attractor, shown in Figure 4c. Additionally, we evaluate the Fourier energy spectrum of the trajectory rollout by the MLP and our model. The comparison between Figure 4b and Figure 4d exemplifies the fact that an unbounded trajectory leads to

unreliable statistics, and that by providing guarantees on generating bounded trajectories, our model is able to preserve the invariant statistics in the true dynamics.

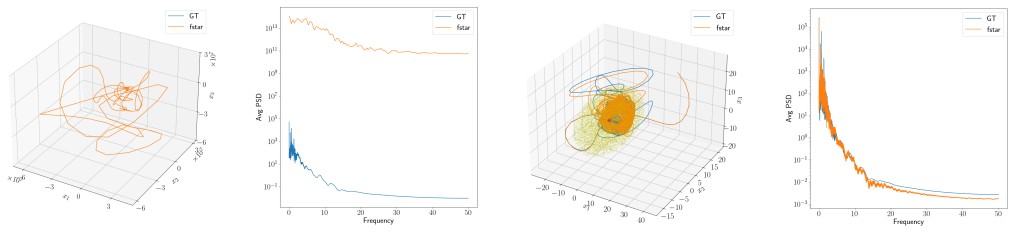

(a) Trajectory rollout with­out projection.  (b) Invalid energy spec­trum after blowup.  (c) Trajectory rollout with projection.  (d) Fourier energy spec­trum.

Figure 4: Lorenz 96 (first 3 states) Vanilla MLP model generates unbounded trajectory (a) and leads to unreliable statistics (b); Our model ensures stability (c) and preserves invariant statistics (d).

### 5.3 TRUNCATED KURAMOTO–SIVASHINSKY EQUATION

We now consider an 8th-order ODE truncated from the KS equation, which is derived in detail in Appendix C. The training dataset consists of 4 different trajectories of 500 steps, sampled at 0.01 [sec] using RK4. A single-step prediction setting is used during training, and both a vanilla MLP and our proposed method based on the same MLP are trained on the same dataset.

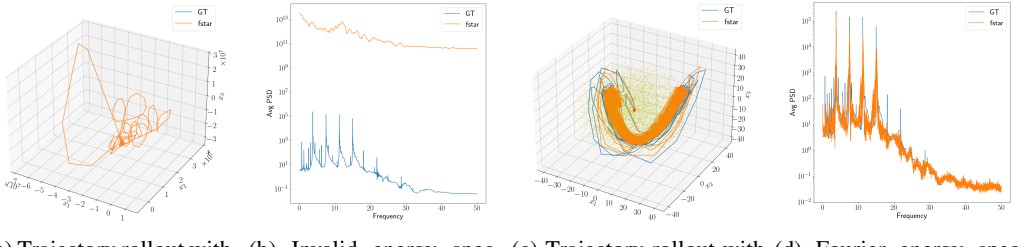

(a) Trajectory rollout with­out projection.  (b) Invalid energy spec­trum after blowup.  (c) Trajectory rollout with projection.  (d) Fourier energy spec­trum.

Figure 5: Truncated KS (first 3 states) Vanilla MLP model generates unbounded trajectory (a) and leads to unreliable statistics (b); Our model ensures stability (c) and preserves invariant statistics (d).

We roll out both models using the RK4 method with a sampling rate of 0.01 [sec] over 50,000 steps. Similar to the observations in the Lorenz 96 example, the vanilla MLP model experiences finite-time blowup (Figure 5a) which leads to invalid energy spectrum evaluation (Figure 5b). With formal guarantees, our model is capable of approximating both the ground truth trajectory long-term behaviors (Figure 5c) and the energy spectrum (Figure 5d) very well.

## 6 CONCLUSION

In this paper, we propose a novel neural network architecture for learning chaotic systems that en­sures the learned dynamics are dissipative, which guarantees the model to always generate bounded trajectories and provide meaningful statistics evaluation. By leveraging control theoretic ideas, we have derived algebraic conditions that ensure dissipativity and embed these conditions into the neural network through a projection layer design. Using the level set of an energy function learned simul­taneously with the dynamics emulator, our model also provides an outer estimate for the strange attractor, which is difficult to characterize due to its complex geometry. Numerical experiments for chaotic systems including Lorenz 96 and truncated KS show the model's capability of preserving invariant statistics of the true dynamics. Experiments also show that a model trained without the stability projection layer leads to finite-time blowup and unreliable statistics, which exemplifies the importance of our proposed approach in learning dissipative chaotic dynamics.

ACKNOWLEDGMENTS

The authors acknowledge the MIT SuperCloud and Lincoln Laboratory Supercomputing Center for providing computing resources that have contributed to the results reported within this paper. This work was supported in part by MathWorks, the MIT-IBM Watson AI Lab, the MIT-Amazon Science Hub, and the MIT-Google Program for Computing Innovation.

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

## A    Proof for Theoretical Results

*Proof for Proposition 1.* Let $x(0) \in M(c)$, suppose there exists $t > 0$ such that $x(t_0) \notin M(c)$, which implies that $V(x(0)) \le c < V(x(t_0))$. Since $V(x(t))$ is continuously differentiable in $t$, by the intermediate value theorem, we can find some $t \in [0, t_0)$ such that $V(x(t_0)) = c$, denote $S = \{t : V(x(t)) = c, t \in [0, t_0)\}$. Since $\forall t \in S, t < t_0$, $\sup S \le t_0$. Suppose $\sup S = t_0$, then we can construct a sequence $(t_k)_{k \in \mathbb{N}}$ such that $t_k \to \sup S$ as $k \to \infty$. By continuity, $V(x(\sup S)) = c$ which contradicts $V(x(t_0)) > c$. Therefore, $\sup S < t_0$. Now by the mean value theorem, there exists $t_1 \in (\sup S, t_0)$ such that $V(x(t_1)) > c$ and $\dot{V}(x(t_1)) = (V(x(t_0)) - V(x(\sup S)))/(t_0 - \sup S) > 0$, which contradicts the assumed condition. Therefore, $\forall t > 0$, we have $V(x(t)) \in M(c)$ as long as $x(0) \in M(c)$, i.e., $M(c)$ is indeed a positively invariant set. $\square$

*Proof for Proposition 2.* Since the condition here is stronger than the one stated in Proposition 1, $M(c)$ is a positively invariant set. Therefore, it suffices to consider a trajectory that starts outside $M(c)$. Suppose there exists a trajectory $x(t)$ such that $\forall t \in [0, \infty), V(x(t)) > c$, then $\dot{V}(x(t)) < 0$ and $V(x(t))$ is monotonically decreasing over time. Since $V(x(t))$ is lower bounded, $V(x(t)) \to a \ge \inf_x V(x)$ as $t \to \infty$. Suppose $a > c$, i.e., $\text{dist}(x(t), M(c)) = \inf_{y \in M(c)} \|y - x(t)\| \not\to 0$ as $t \to \infty$.

Since $V$ is radially unbounded, for any $\alpha > 0$, we can find $r_\alpha$ such that $V(x) > \alpha$ for all $\|x\| > r_\alpha$. Therefore, any level set of $V$ is bounded as $\{x : V(x) \le \alpha\} \subset B(r_\alpha)$. Note that $V(x(t)) \in [a, V(x(0))]$ for all $t \in [0, \infty)$ because $V(x(t))$ is monotonically decreasing. Since $V(x)$ is continuous, the pre-image $S = \{x : V(x) \in [a, V(x(0))]\}$ is a closed set. Additionally, $S$ is bounded because $S \subset \{x : V(x) \le \alpha\} \subset B(r_{V(x(0))})$, which implies $S$ is compact.

Since $\dot{V}(x)$ is continuous and $\dot{V}(x) < 0$ for all $x \in S$, there is $\gamma > 0$ such that $\max_{x \in S} \dot{V}(x) \le -\gamma$, which implies $\max_{t \in [0, \infty)} \dot{V}(x(t)) \le \max_{\|x\| < r_{x(0)}} \dot{V}(x) \le -\gamma$. This contradicts the fact that $V(x(t)) \ge a > -\infty$ for all $t \in [0, \infty)$ since

$$V(x(t)) = V(x(0)) + \int_0^t \dot{V}(x(\tau))d\tau \le V(x(0)) - \gamma t.$$

$\square$

*Proof for Corollary 1.* If $\frac{\partial V}{\partial x} \hat{f}(x) + V(x) - c \le 0$, it follows from (5) that $f^*(x) = \hat{f}(x)$. Therefore, in this case, $\frac{\partial V}{\partial x} f^*(x) + V(x) - c \le 0$, meaning that $f^*(x)$ satisfies the condition in (3).

If $\frac{\partial V}{\partial x} \hat{f}(x) + V(x) - c > 0$, then we have the following,

$$\frac{\partial V}{\partial x} f^*(x) + V(x) - c = \frac{\partial V}{\partial x} \hat{f}(x) - \frac{\partial V}{\partial x} \frac{\partial V}{\partial x}^T \frac{\text{ReLU}\left(\frac{\partial V}{\partial x} \hat{f}(x) + V(x) - c\right)}{\|\frac{\partial V}{\partial x}\|^2} + V(x) - c$$

$$= \frac{\partial V}{\partial x} \hat{f}(x) - \left(\frac{\partial V}{\partial x} \hat{f}(x) + V(x) - c\right) + V(x) - c = 0$$

which ensures that $f^*(x)$ satisfies the stability condition (3). Therefore, by Theorem 1, the system $\dot{x} = f^*(x)$ is dissipative and $M(c)$ is globally asymptotic stable.

Additionally, since $\forall x \notin M(c), \dot{V}(x) < -(V(x) - c) < 0$, the trajectory always lose energy outside $M(c)$. Also note that if the trajectory starts within $M(c)$, it can never leave, i.e., $V(x(t)) \le c$ for all $t \ge 0$. Therefore, for any $t \in [0, \infty)$, $V(x(t)) = V(x(0)) + \int_0^t \dot{V}(x(\tau))d\tau \le \max\{V(x(0)), c\}$. Since every level set of $V(x(t))$ is bounded (see proof for Proposition 2), $x(t)$ is always bounded. $\square$

## B    Additional Numerical Experiment Results

### B.1    Statistics Evaluation

In Section 5, we have presented a few cases where an unconstrained MLP could lead to finite blowup when generating predicted trajectories for Lorenz 96 and truncated KS systems, which makes the

statistics evaluation unreliable. Additionally, we have shown examples where our model is able to produce bounded trajectories that preserve the invariant statistics in the true system, e.g., in Figure 4d and 5d. Here we provide more quantitative evidence that validates our model's guarantees in generating bounded trajectories and its capability of preserving invariant statistics, specifically the Fourier energy spectrum. We simulate 25 trajectories of 50000 steps with a sampling rate $h = 0.01$ [sec] using both an unconstrained MLP and our proposed model with stability projection. From these 25 trajectories, we evaluate the average Fourier energy spectrum, and compute the percentage errors when compared to the Fourier energy spectrum obtained from trajectories simulated using true dynamics. The results are summarized in the following table:

|  | MLP error % | MLP #Unbounded Trajs | Proposed Model error % |
|---|---|---|---|
| Lorenz 63 | $6.37 \times 10^{-4}$ | 0 | $7.40 \times 10^{-4}$ |
| Lorenz 96 | N/A | 24 | $1.96 \times 10^{-3}$ |
| Truncated KS | N/A | 6 | $3.37 \times 10^{-3}$ |

Table 1: Fourier Energy Spectrum Percentage Error

As shown in Table 1, our proposed model achieves very low percentage error in producing the Fourier energy spectrum, which is an important invariant statistics in dissipative chaotic systems. It is also observed that unconstrained MLP experiences finite blow-up quite often: Out of 25 trajectories, 24 grow unbounded in the Lorenz 96 example and 6 grow unbounded in the truncated KS example, both rendering the statistics evaluation invalid as NaN values appear. Note that Lorenz 63 is intrinsically much more dissipative than the other two examples, which could be part of the reason why an unconstrained model does not experience stability issues. Nonetheless, our proposed model achieves comparable performance with the unconstrained MLP in this case as well.

## B.2 LEARNED LYAPUNOV FUNCTION AND DISSIPATIVITY

To visualize the learned Lyapunov function and its connection to dissipative behaviors in the dynamics, we plot the learned energy function evaluated at two trajectories, one generated by the true dynamics and the other generated by our model. More specifically, starting from the same initial condition $x(0) = x^*(0)$, we forward simulate the true dynamics $\dot{x} = f(x)$ and the learned model $\dot{x}^* = f^*(x^*)$ and obtain two trajectories $x(t)$ and $x^*(t)$, respectively. The trajectories are numerically integrated using RK4 methods for 50000 steps with a sampling period of $h = 0.01$ [sec]. We present the comparison between $V(x(t))$ and $V(x^*(t))$, the energy level time evolution for the true and learned dynamics with Lorenz 63, Lorenz 96, truncated KS in Figure 6a, 6b, 6c, respectively.

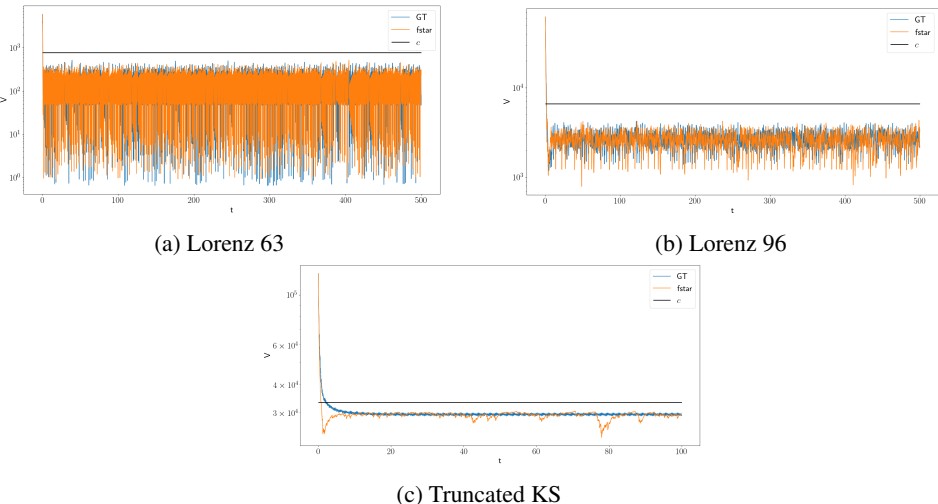

(a) Lorenz 63      (b) Lorenz 96

(c) Truncated KS

Figure 6: Lyapunov function time histories of true and learned dynamics trajectory rollouts.

As shown in Figure 6, in all three examples, the learned dynamics trajectory rollout energy level matches with that of real dynamics very well. Since the goal is not to perform accurate pointwise prediction, from a perspective of matching statistics, two trajectories share very similar range and trends in energy time histories. Additionally, the energy of all trajectories all start above the level set parameter $c$, and quickly loses energy but continues to stay active to gain or lose energy within the level set $M(c) = \{x : V(x) \leq c\}$, which matches with the dissipative behavior of these systems. It is also observed that the maximum energy level after trajectories enter $M(c)$ stays below while close to $c$, which validates the fact that the learned level sets are reasonably tight outer estimates for the strange attractors.

### B.3 PROJECTED 2D TRAJECTORY VISUALIZATIONS

In addition to the 3D plots presented in Section 5, we provide their 2D projections to better visualize the trajectory behavior over the state space and the learned invariant level set. In Figure 7, 8 and 9, we present the true and predicted trajectory and the invariant level set (ellipsoid sample points in yellow) projected in $x_1 - x_2, x_2 - x_3, x_3 - x_1$ 2D planes for Lorenz 63, Lorenz 96, and the truncated KS system, respectively. These plots further illustrate the fact that our model generates bounded trajectories which recover the strange attractor, and the learned level sets are reasonably tight outer estimates for the strange attractors.

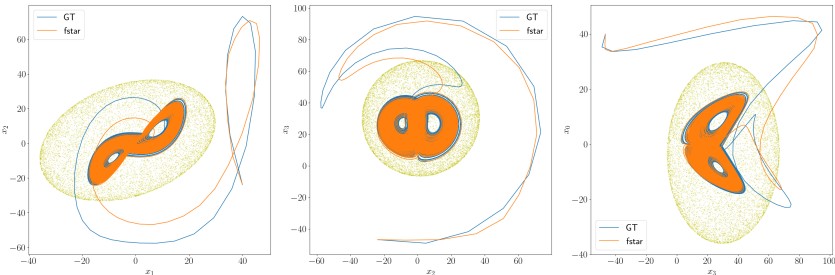

Figure 7: Lorenz 63: Trajectory and Invariant Set projected to 2D planes.

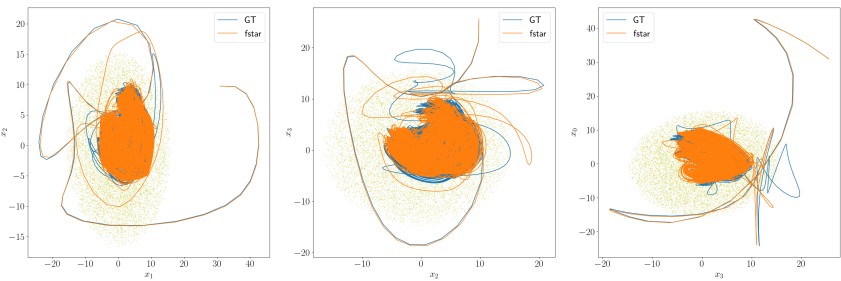

Figure 8: Lorenz 96: Trajectory and Invariant Set (in first 3 states) projected to 2D planes.

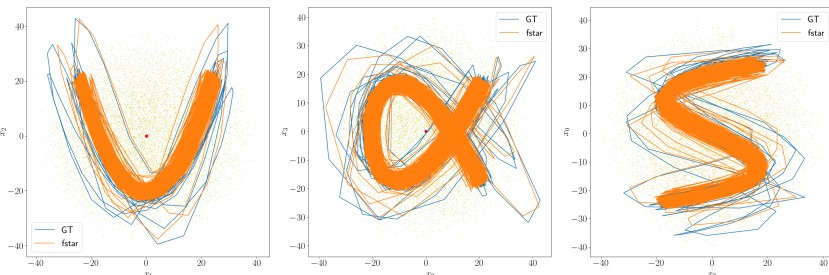

Figure 9: Truncated KS: Trajectory and Invariant Set (in first 3 states) projected to 2D planes.

### B.4 TRAINING LOSS HISTORY

To provide clarifications into the impact of the stability projection layer on the training process, we include the training loss history over epochs in Figure 10. The dynamic loss refers to the trajectory prediction loss $\frac{1}{NT}\sum_{i=1}^{N}\sum_{k=1}^{T}\|x_k^{(i)} - \hat{x}_k^{(i)}\|_2^2$, and the regularization loss refers to $\lambda \text{Vol}(M(c))$, which are introduced in the loss function definition (6). As observed in every case presented in Figure 10, the stability projection layer does not add difficulties to training loss convergence.

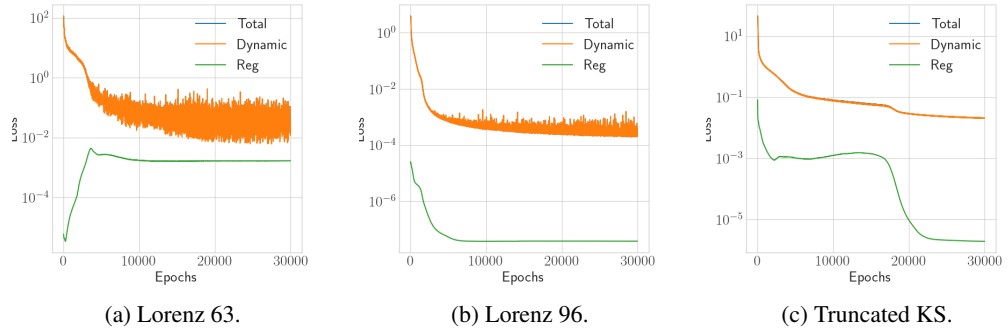

(a) Lorenz 63.  (b) Lorenz 96.  (c) Truncated KS.

Figure 10: Training loss history over epochs.

## C TRUNCATED ODE APPROXIMATION FOR KS PDE

Consider the scaled KS equation,

$$u_t + uu_x + u_{xx} + \nu u_{xxxx} = 0 \tag{7a}$$
$$(x, t) \in \mathbb{R} \times \mathbb{R}^+ \tag{7b}$$
$$u(x, t) = u(x + 2\pi, t) \tag{7c}$$

where the last equation represents the periodic boundary condition, and $\nu = \frac{\pi^2}{L^2}$ represents a dimensionless wavelength (in other versions of KS equations, the boundary conditions typically involve a wavelength $L$). In general, smaller $\nu$ corresponds to more chaotic, less dissipative behaviors, as $\nu \to 0$ corresponds to wavelength becoming infinitely long.

Following the derivation in (Smyrlis & Papageorgiou, 1996), we apply the Galerkin projection method (Holmes, 2012) to acquire a finite-dimensional truncated ODE that approximates the KS equation. Since the solution is spatially periodic in $2\pi$ as stated in (7c), the solution $u(x, t)$ can be represented using Fourier basis as follows:

$$u(x, t) = \sum_{k \geq 1}(\alpha_k(t)\cos kx + \beta_k(t)\sin kx) + \alpha_0(t) \tag{8}$$

Note that the periodic condition (7c) suggests conservation of energy in the KS equation. More specifically, (7c) implies that

$$\int_0^{2\pi}(uu_x + u_{xx} + \nu u_{xxxx})dx = \left[\frac{u^2}{2} + u_x + \nu u_{xxx}\right]_0^{2\pi} = 0$$

$$\implies \int_0^{2\pi} u_t dx = -\int_0^{2\pi}(uu_x + u_{xx} + \nu u_{xxxx})dx = 0$$

$$\implies \frac{d\alpha_0(t)}{dt} = \frac{1}{2\pi}\frac{d}{dt}\int_0^{2\pi} u(x, t)dx = \frac{1}{2\pi}\int_0^{2\pi} u_t(x, t)dt = 0$$

which means $\alpha_0(t)$ is constant. Without loss of generality, let us assume $\alpha_0(t) = 0$, and rewrite the PDE in (7a) with its Fourier basis representation (8). As a result, we obtain the following infinite-

dimensional ODE from the PDE,

$$\alpha'_k = \lambda_k \alpha_k + A_k \tag{9a}$$

$$\beta'_k = \lambda_k \beta_k + B_k \tag{9b}$$

where $k \in \mathbb{N}$, $\lambda_k = k^2 - \nu k^4$, and

$$A_k = -\frac{k}{2} \sum_{m+n=k} \alpha_m \beta_n + \frac{k}{2} \sum_{m-n=k} (\alpha_m \beta_n - \alpha_n \beta_m) \tag{10a}$$

$$B_k = \frac{k}{4} \sum_{m+n=k} (\alpha_m \alpha_n - \beta_m \beta_n) + \frac{k}{2} \sum_{m-n=k} (\alpha_m \alpha_n + \beta_m \beta_n) \tag{10b}$$

We can now obtain a finite-dimensional truncated system of ODEs by only considering a few Fourier bases, $k = 1, 2, ..., N$. The resulting truncated ODEs can be written as

$$\alpha'_k = \lambda_k \alpha_k + A_k^N(\alpha_1, ..., \alpha_N, \beta_1, ..., \beta_N), \quad k = 1, 2, ..., N \tag{11a}$$

$$\beta'_k = \lambda_k \beta_k + B_k^N(\alpha_1, ..., \alpha_N, \beta_1, ..., \beta_N), \quad k = 1, 2, ..., N \tag{11b}$$

$$A_k^N = -\frac{k}{2} \sum_{m=1}^{k-1} \alpha_m \beta_{k-m} + \frac{k}{2} \sum_{m=1}^{N-k} (\alpha_{m+k}\beta_m - \alpha_m \beta_{m+k}) \tag{11c}$$

$$B_k^N = \frac{k}{4} \sum_{m=1}^{k-1} (\alpha_m \alpha_{k-m} - \beta_m \beta_{k-m}) + \frac{k}{2} \sum_{m=1}^{N-k} (\alpha_{m+k}\alpha_m + \beta_{m+k}\beta_m) \tag{11d}$$

To reduce the computational cost of learning from trajectory data generated by the truncated ODEs, we choose $N = 4$ to conduct numerical experiments. The truncated 8-dimensional ODE still preserves chaotic behaviors, as reported in (Smyrlis & Papageorgiou, 1996).

