# OpenReview forum: "Learning Chaotic Dynamics with Embedded Dissipativity"
_ICLR.cc/2025/Conference — ICLR 2025 Conference Withdrawn Submission_

### Official Review · Reviewer_7Bsh · 2024-10-28

**Soundness:** 2
**Presentation:** 3
**Contribution:** 2
**Rating:** 3
**Confidence:** 3

**Summary:**

The paper introduces a novel neural network architecture to model chaotic systems with dissipative properties. The proposed model uses Lyapunov control-theoretic methods to ensure that the system maintains dissipative behaviours, effectively providing guarantees on the boundedness of the attractors. The proposed architecture, which includes a stability projection layer, was tested on systems was tested on some standard chaotic systems. This result significantly improves models without the stability projection layer, which is prone to unbounded errors. These results contribute to more reliable learning of chaotic dynamics.

**Strengths:**

The targeted problem is highly significant in chaotic systems modelling, and the general idea is promising. The proposed approach effectively integrates control-theoretic concepts with machine learning to ensure bounded trajectory generation, contributing to the dissipative properties of chaotic systems—an essential aspect for the validity of invariant statistics. The paper is well-structured and presents the proposed method concisely.

**Weaknesses:**

## Experiments
- The numerical experiment lacks a comparison with updated approaches in chaotic system modelling, limiting the perspective on how much improvement the proposed model offers.
- The systems examined in the experiments appear simple and low-dimensional (no more than five states). This raises concerns about the scalability of the proposed method.
- The weight hyperparameter $\lambda$ should be critical in balancing forecasting accuracy with estimating the attraction region. However, an ablation study on the impact of $\lambda$ is missing. The robustness of the initial choice of $c$ should also be evaluated.
- While the proposed method successfully generates bounded trajectories, the investigation into long-term prediction accuracy and related statistics is missing in the numerical experiments. The provided Figures 3, 4, and 5 are significantly insufficient for verifying the effectiveness of the method. Boundedness alone does not ensure the precise or unbiased estimation of long-term statistical properties.

## Some minor issues
- Typos "[sec]" in Line 388,395,397,441,505.
- Line 396, "we *forward simulate* the learned ..."

 I’d encourage the authors to redo a proofread.

**Questions:**

- The authors claim to learn an outer estimate of the complex strange attractor. However, the proposed method utilizes a simple ellipsoid to approximate the strange attractor without providing any quantitative results to support this claim. Could the authors provide a more detailed justification for why an ellipsoid is an appropriate representation of complex attractors and include quantitative verifications?
- Propositions 1 and 2 appear to be well-known results derived from Lyapunov's Direct Method [Chapter 3, Slotine et al. (1991)]. Could the authors clarify how these propositions differ from existing results? Why were appropriate citations not provided if they do not offer novel contributions?

References

- Slotine, Jean-Jacques E., and Weiping Li. Applied nonlinear control. Vol. 199. No. 1. Englewood Cliffs, NJ: Prentice hall, 1991.

---

> ### Author Response · Authors · 2024-11-16
> **We thank the reviewer for their comments and provided responses (Part I)**
>
> Format: *original review comments* followed by responses
> **Weaknesses:**
> **Experiments**
> 1. *The numerical experiment lacks a comparison with updated approaches in chaotic system modelling, limiting the perspective on how much improvement the proposed model offers.*
>
> - We thank the reviewer for this comment, and would like to provide clarifications for why we think the current experimetal setup which compares the same model with stability projection on (our model) and off (vanilla MLP) is appropriate for validating our approach as follows:
>     - As allueded to in the paragraph above related work, which is part of the literature review, sequential time-series models including RNN and RC-based approaches have been shown to generate unbounded trajectory predictions. The goal of our paper is to propose a method that provides boundedness guarantees, not necessarily with improved prediction performance over current benchmarks in the cases where they are able to generate bounded trajectories. In the case where a model developed in an updated approach in the literature generate unbounded trajectory predictions, finite-time blow-up will lead to unreliable statistics the same way as reported in the vanilla MLP case (Fig 4/5 (a)(b)) as trajectory value will grow unrealistically large.
>     - the current experiments compare the same model with the stability projection layer on (our model) and off (vanilla MLP), which serves as an ablation study that investigates the effect of the stability projection by training two models on the exact same dataset. Additionally, other methods without stability guarantees that include either sequential models e.g. (Platt et al. 2023) or statistics matching training procedures e.g. (Jiang et al. 2024) may not be fair, as they have access to more information during training than our methods which can only access trajectory segments.
>
>
> 2. *The systems examined in the experiments appear simple and low-dimensional (no more than five states). This raises concerns about the scalability of the proposed method.*
> - We thank the reviewer for the scability concern comment. As the reviewer suggests, we are working on higher dimension examples e.g., 2D Kolmogorov flow on a 16-by-16 grids, to validate the scalability of our method, and we will include these additional examples in the camera-ready version.
> - We would like to clarify that the truncated KS system is an 8-dimensional ODE that observes chaotic behaviors.
>
> 3. *The weight hyperparameter λ should be critical in balancing forecasting accuracy with estimating the attraction region. However, an ablation study on the impact of λ is missing. The robustness of the initial choice of c should also be evaluated.*
> - We thank the reviewer for this comment. We will include an ablation study that shows the impact of $\lambda$.
> - Additionally, the strategy of choosing a sufficiently large $c$ works well in practice. Intuitively, using a very large initial $c$ effectively allows the network to learn without projection, and only enforces projection later on when $c$ is reduced due to the regularization loss.
>
> 4. *While the proposed method successfully generates bounded trajectories, the investigation into long-term prediction accuracy and related statistics is missing in the numerical experiments. The provided Figures 3, 4, and 5 are significantly insufficient for verifying the effectiveness of the method. Boundedness alone does not ensure the precise or unbiased estimation of long-term statistical properties.*
>
> - We agree with the reviewer that more statistical metrics can be helpful.we are working on providing additional statistics evaluations and will include them in the camera-ready version to improve performance evaluation.
> - We would like the clarify the problem setup. As alluded to in the literature review, many models suffer from not having a trajectory boundedness guarantee and are known to generate unbounded trajectories. This is a crucial problem because as shown in Figure 4 and 5, if the system generates an unbounded trajectory, the statistics evaluation e.g., energy spectrum is rendered unreliable.
> - The goal of our paper is to propose a method that provides boundedness guarantees. Therefore, the current experiments that compare models with stability projection layer on and off can validate the proposed approach.
>
>
> **Some minor issues**
> *Typos "[sec]" in Line 388,395,397,441,505.
> Line 396, "we forward simulate the learned ..."
> I’d encourage the authors to redo a proofread.*
>
> - We thank the reviewer for carefully reading through our paper and the detailed feedback. We would like to clarify that “[sec]” is a standard notation for the unit second, and that “forward simulate” means simulate the learned ODE forward in time, which is a standard notion in the dynamical system literature.

---

> > ### Author Response · Authors · 2024-11-16
> > **Continued responses**
> >
> > **Questions:**
> > 1. *The authors claim to learn an outer estimate of the complex strange attractor. However, the proposed method utilizes a simple ellipsoid to approximate the strange attractor without providing any quantitative results to support this claim. Could the authors provide a more detailed justification for why an ellipsoid is an appropriate representation of complex attractors and include quantitative verifications?*
> >
> > - While this reviewer correctly mentions “outer estimate”, the level set ellipsoid is not an approximate to the strange attractor, i.e., we are not attempting to learn an accurate representation or shape of the attractor.
> > - Similar to common approaches in the literature (e.g., Li et al. 2022), during testing, we can reconstruct an approximation to the strange attractor from the predicted trajectories empirically. This is also achieved in the numerical examples, specifically, our model predicted a trajectory that recovers the strange attractor in all three cases.
> > - Additionally, we would like to clarify that the effect of the Lyapunov function is only to guide the trajectory to asymptotically converge to the level set that contains the attractor. It is true that chaotic systems observe complex behaviors, but these behaviors (as discussed thoroughly in Section 2) almost always occur on the strange attractor but not elsewhere for dissipative systems. Since the Lyapunov function has no effect after the trajectory enters the level set, and since the attractor is inside the level set, the ReLU function that incorporates the Lyapunov function information will be inactive thus has little or no influence on learning complex chaotic behaviors.
> >
> > 2. *Propositions 1 and 2 appear to be well-known results derived from Lyapunov's Direct Method [Chapter 3, Slotine et al. (1991)]. Could the authors clarify how these propositions differ from existing results? Why were appropriate citations not provided if they do not offer novel contributions?*
> >
> > - Despite similarity in appearance, standard Lyapunov theory including the book chapter mentioned above addresses invariant principle and asymptotic stability with respect to an equilibrium point. Here, Proposition 1 and 2 are derived to describe asymptotic convergence and invariance of a level set, rather than an equilibrium point. Although we use similar real analysis techniques to establish the proof, these results are technically different than the existed ones in literature.
> > - Additionally, we cited a well-known nonlinear system textbook that contains theoretical results and analysis techniques necessary for our propositions, thus provide attributes to this known line of study. Specifically, the textbook is “Nonlinear Systems” by Hassan Khalil.
> >
> > References
> > Slotine, Jean-Jacques E., and Weiping Li. Applied nonlinear control. Vol. 199. No. 1. Englewood Cliffs, NJ: Prentice hall, 1991.

---

> > > ### Comment · Reviewer_7Bsh · 2024-11-24
> > >
> > > Thank you for your detailed response, which clarified several points. However, three concerns remain:
> > >
> > > - While the authors emphasize boundedness as the primary goal of this work, I question whether it is **sufficient** as the key property in learning dynamical systems. From the reviewer’s perspective, boundedness is helpful but should not come at the cost of accuracy in short-term predictions or long-term statistical properties, which are central objectives in learning dynamical systems. Providing empirical evidence that the proposed method achieves comparable accuracy to existing approaches, such as reservoir computing (Pathak, et al., 2018; Yan, et al., 2024), which has demonstrated effectiveness in learning chaotic systems, would strengthen the work.
> > >
> > > - Since the focus is on boundedness, it is reasonable to include comparisons with at least one of the Li et al., (2022) and Schiff et al., (2024) that also emphasize learning dissipative chaotic systems . Could the authors clarify why these comparisons were not conducted or included? While I know the hyperparameter challenges in (Li et al., 2022) as mentioned in the introduction, this method could still be at least included for the Lorenz-63 system, where reproducible code and tuned results are available.
> > >
> > > - The scalability issue remains unresolved. Upscaling to a 16×16 grid does not adequately address this issue, as existing methods operate at much higher spatial resolutions. Demonstrating competitive results at a scale comparable to that shown in  Li et al., (2022) or Schiff et al., (2024) would address this concern.
> > >
> > > Finally, the concept of learning the level-set of the Lyapunov function is an interesting direction to modelling dissipative chaotic systems, offering theoretical insights as demonstrated by the authors. However, the lack of a solid empirical investigation and potential solutions leaves key concerns unresolved. Therefore, my scores remain based on the three concerns outlined above.
> > >
> > >
> > > - References
> > >   - Yan, M., Huang, C., Bienstman, P. et al. Emerging opportunities and challenges for the future of reservoir computing. Nat Commun 15, 2056 (2024). https://doi.org/10.1038/s41467-024-45187-1
> > >
> > >   - Pathak, J., Hunt, B., Girvan, M., Lu, Z., & Ott, E. (2018). Model-Free Prediction of Large Spatiotemporally Chaotic Systems from Data: A Reservoir Computing Approach. Physical Review Letters, 120(2), 024102.
> > >
> > >   - Li, Zongyi, et al. "Learning dissipative dynamics in chaotic systems." Proceedings of the 36th International Conference on Neural Information Processing Systems. 2022.
> > >
> > >   - Schiff, Yair, et al. "DySLIM: Dynamics Stable Learning by Invariant Measure for Chaotic Systems." Forty-first International Conference on Machine Learning. 2024.

---

> > > > ### Author Response · Authors · 2024-12-01
> > > > **Responses to Reviewer comments**
> > > >
> > > > We thank the reviewer for elaborating their concerns. We acknowledge the importance of higher-dimensional numerical examples and will include those in the next iteration of the paper. In addition, we hope the following responses clarify some of the reviewer's concerns.
> > > >
> > > > 1. As discussed in the related work section, most methods in the literature including reservoir computing based methods are still prone to unbounded trajectory generation, or finite-time blow-up, which leads to unreliable statistics evaluation or reduced prediction horizon. Therefore, our proposed approach which provides formal trajectory boundedness guarantees is a significant advancement and addresses an important open problem. While comparison of prediction accuracy is an important metric, the absence of accuracy results does not diminish theoretical contributions.
> > > > 2. The distinctions between our method and other approaches that also encourage dissipativity are two-folds: (1) Our method provides theoretical boundedness guarantees while other methods have no formal guarantee (2) Our method provides a new perspective on leveraging control theory to embed dissipative conditions as an inductive bias of the model. The numerical experiments in our current paper aim to compare our proposed approach with and without the stability projection layer. Such a comparison highlights the benefits of building a model with trajectory boundedness guarantees. Comparing against, e.g., (Li et al. 2022) does not seem to add too much values, since the current experiments well establish the importance of obtaining trajectory boundedness guarantees.
> > > > 3. The goal of this paper is to propose a predictive model for finite-dimensional chaotic systems that is inherently dissipative. Despite the practical importance of PDEs like Kuramoto-Sivashinsky or Kolmogorov flow, we do not claim that our framework directly addresses learning PDE solution operators like the problem settings considered in (Li et al. 2022) and (Schiff et al. 2024). The requirement that the framework should be scalable to produce dissipative ODE models that are of very high dimensions effectively approximating PDEs may not be suitable for the ODE learning setting under our consideration.

---

### Official Review · Reviewer_Y8Lv · 2024-10-30

**Soundness:** 2
**Presentation:** 3
**Contribution:** 2
**Rating:** 5
**Confidence:** 4

**Summary:**

This paper focuses on learning models to predict chaotic systems’ behaviour while maintaining dissipative properties. It addresses the challenge of achieving long-term stability in chaotic system predictions by proposing a neural network model with embedded dissipativity. This model incorporates Lyapunov function principles from control theory to ensure that generated trajectories remain bounded, thus avoiding statistical invalidation due to unbounded trajectories.

Although the idea of "embedding dissipativty" has been mentioned in some literature combining deep learning and control, the unique contribution of this work is to introduce algebraic conditions combined with a stability projection layer, providing an efficient implementation method. This projection layer is based on the Lyapunov method and performs energy constraints in the neural network, avoiding the need to rely on external data or expert knowledge.

**Strengths:**

The concept of dissipative embedding has been explored in related fields, this paper proposes an innovative approach in terms of specific implementation methods. It introduces Lyapunov functions into the neural network structure and ensures that the generated trajectory satisfies dissipativity through a stability projection layer, thereby keeping the trajectory bounded in the long-term prediction of chaotic systems. This approach of combining the stability conditions of control theory with deep learning models provides a new perspective for modeling chaotic systems and reduces the reliance on empirical regularization.

The paper is written with high clarity and readability and is able to combine complex control theory concepts with neural network models well, allowing readers to understand its theoretical motivation and implementation methods. The author provides intuitive explanations when introducing concepts such as Lyapunov stability, dissipation, and strange attractors, and enhances understanding through diagrams and formulas. The overall structure is logically clear, and the paper has a good connection from problem definition, and method design to experimental verification.

**Weaknesses:**

The paper lacks citations to related work in some areas, e.g., Markov neural operator (MNO) in the NeurIPS 2022 paper "Learning Dissipative Dynamics in Chaotic Systems". If its method can be compared with existing research in more detail, especially in the field of dissipative embedding, readers will have a more comprehensive understanding of the innovativeness of this method.

The experiment part of this paper is insufficient for the following reasons:
- No comparison with benchmark algorithms: There are some mature benchmark algorithms in the field of long-term prediction of chaotic systems, such as Reservoir Computing (RC) and Recurrent Neural Networks (RNN) (especially LSTM and GRU), etc. It is recommended that the author select these algorithms and compare them with their methods under the same experimental environment and initial conditions to demonstrate the advantages of the new method in long-term stability and dissipation.
- Lack of statistical performance evaluation metrics: Introduce commonly used chaotic system evaluation metrics, such as Lyapunov exponent (used to quantify sensitivity to initial conditions), attractor reconstruction error, and autocorrelation, and compare the performance of the new method and the benchmark algorithm on these metrics. This will help verify the effectiveness of the paper's method in preserving the statistical properties of chaotic systems.
- Lack of testing on high-dimensional chaotic systems: In addition to the existing Lorenz system and Kuramoto-Sivashinsky equation, it is recommended to add other chaotic systems (such as Rossler system or higher-dimensional Navier-Stokes system) for testing to prove the versatility of the method. This can demonstrate the adaptability and stability of the new method under different complexities and dimensions.
- No ablation experiment: In order to gain a deeper understanding of the role of the stability projection layer, an ablation experiment can be designed. For example, the stability projection layer can be removed from the model to observe the changes in its stability in long-term predictions. This will be able to specifically show the impact of embedding dissipative structures on model performance, thereby further verifying the necessity of this innovation.

**Questions:**

1. Selection and generalization of Lyapunov functions: Although Lyapunov functions are very useful tools in control theory, it may not be easy to design and select appropriate Lyapunov functions in complex chaotic systems, especially in multidimensional or nonlinear systems. Many times, the form of the Lyapunov function needs to be manually designed or assumed based on the specific system, and may lack generalization. Is this approach applicable to a wider range of chaotic systems? In complex practical applications, finding a suitable Lyapunov function may require a lot of expert knowledge and manual design, which limits scalability. If the choice of Lyapunov function is not robust enough, the model may not be applicable to different types of chaotic systems.

2. Design complexity and effectiveness of the projection layer: The projection layer is designed to ensure that the learned dynamic simulator is dissipative, which means that some parts of the neural network are explicitly designed to obey specific constraints (such as trajectory boundedness). However, the design and implementation of the projection layer can be complex. The specific implementation of the projection layer in the neural network and how to ensure that it does not over-constrain the model flexibility is a key issue. The projection layer may affect the learning ability of the neural network, especially when the model needs to deal with different types of nonlinear chaotic systems. Excessive projection may limit the expressive power of the model, causing it to fail to capture some more complex dynamic behaviors.

3. The paper claims to parameterize the Lyapunov function V as a quadratic function for simplicity and points out that the framework can accommodate any function that satisfies the requirements of Theorem 1. However, this statement is reasonable in theory but may have limitations in practical applications.
- In practical applications, non-quadratic Lyapunov functions often significantly increase computational complexity. For example, parameterized as a neural network or a high-order polynomial will bring additional difficulties in gradient calculation and projection layer implementation.
- Complex Lyapunov functions may be more difficult to maintain the required monotonicity and positive definiteness in the model, especially in high-dimensional chaotic systems.
- Complex V forms may lead to numerical instability, especially in the training process, which is prone to gradient explosion or disappearance.
- Suggestions for improvement: If the authors hope to actually expand to a more general Lyapunov function form, the author can consider providing several experimental verifications of non-quadratic parameterized V to demonstrate the effectiveness of the framework in these cases. This work can further analyze the trade-offs between different V forms in terms of numerical stability and computational complexity to demonstrate the practical applicability and versatility of the framework.

---

> ### Author Response · Authors · 2024-11-16
> **We thank the reviewer for their detailed comments and provide responses (Part I)**
>
> (format *original review comments* followed by response)
> **Weaknesses:**
> 1. *The paper lacks citations to related work in some areas, e.g., Markov neural operator (MNO) in the NeurIPS 2022 paper "Learning Dissipative Dynamics in Chaotic Systems". If its method can be compared with existing research in more detail, especially in the field of dissipative embedding, readers will have a more comprehensive understanding of the innovativeness of this method.*
>
> - We would like to respectfully remind the reviewer that the MNO paper has been cited (Li et al. 2022) and discussed in the Related work section on page 2.
> - More importantly, to our best understanding of the literature as discussed in the related work section, most methods do not provide theoretical guarantee for trajectory boundedness, which is crucial for obtaining predicted trajectories with meaningful statistics.
> - Despite the similar notions used in our paper and the MNO paper, our paper provides a different perspective that leverages control theory to learn the dynamics and the bounded region simultaneously, rather than the MNO approach which uses a manually picked ball to encourage dissipativity. Additionally, the MNO paper enforces dissipativity in a pre-defined region by attaching a post-processing layer to the trained network only during testing. In comparison, our method uses active projection both during training and testing, which avoids the requirement in the MNO paper to pre-process the trajectory data into transient and ergodic phases, thus making our method more flexible to be applicable to any trajectory data.
>
> 2. *The experiment part of this paper is insufficient for the following reasons:*
>     1. *No comparison with benchmark algorithms: There are some mature benchmark algorithms in the field of long-term prediction of chaotic systems, such as Reservoir Computing (RC) and Recurrent Neural Networks (RNN) (especially LSTM and GRU), etc. It is recommended that the author select these algorithms and compare them with their methods under the same experimental environment and initial conditions to demonstrate the advantages of the new method in long-term stability and dissipation.*
>
>     - As allueded to in the paragraph above related work, which is part of the literature review, sequential time-series models including RNN and RC-based approaches have been shown to generate unbounded trajectory predictions. The goal of our paper is to propose a method that provides boundedness guarantees, not necessarily with improved prediction performance over current benchmarks in the cases where they are able to generate bounded trajectories. In the case where they generate unbounded trajectory predictions, finite-time blow-up will lead to unreliable statistics same as reported in the vanilla MLP case (Fig 4/5 (a)(b)) as trajectory value will grow unrealistically large.
>     - We would like to point out, the current experiments compare the same model with the stability projection layer on (our model) and off (vanilla MLP), which also serves as an ablation study that investigates the effect of the stability projection by training two models on the exact same dataset. Additionally, other methods without stability guarantees that include either sequential models e.g. (Platt et al. 2023) or statistics matching training procedures e.g. (Jiang et al. 2024) may not be fair, as they have access to more information during training than our methods which can only access trajectory segments.
>
>     2. *Lack of statistical performance evaluation metrics: Introduce commonly used chaotic system evaluation metrics, such as Lyapunov exponent (used to quantify sensitivity to initial conditions), attractor reconstruction error, and autocorrelation, and compare the performance of the new method and the benchmark algorithm on these metrics. This will help verify the effectiveness of the paper's method in preserving the statistical properties of chaotic systems.*
>
>     - We thank the reviewer for this comment. As the reviewer suggests, we are working on providing additional statistics evaluations and will include them in the camera-ready version to improve performance evaluation.
>
>     3. *Lack of testing on high-dimensional chaotic systems: In addition to the existing Lorenz system and Kuramoto-Sivashinsky equation, it is recommended to add other chaotic systems (such as Rossler system or higher-dimensional Navier-Stokes system) for testing to prove the versatility of the method. This can demonstrate the adaptability and stability of the new method under different complexities and dimensions.*
>     - We thank the reviewer for this comment. As the reviewer suggests, we are working on higher dimension examples e.g., 2D Kolmogorov flow on a 16-by-16 grids, to validate the scalability of our method, and we will include these additional examples in the camera-ready version.

---

> > ### Author Response · Authors · 2024-11-16
> > **Continued Responses**
> >
> > **Weaknesses (cont'd)**
> > 2-4. *No ablation experiment: In order to gain a deeper understanding of the role of the stability projection layer, an ablation experiment can be designed. For example, the stability projection layer can be removed from the model to observe the changes in its stability in long-term predictions. This will be able to specifically show the impact of embedding dissipative structures on model performance, thereby further verifying the necessity of this innovation.*
> >     - We agree that direct comparison between with and without stability projection is effective in showing the impact of our approach in guaranteeing trajectory boundedness. We respectfully remind the reviewer that this is exactly the comparison setting in Figure 4 and 5. The difference between (a)(b) and (c)(d) are that (a)(b) corresponds to the prediction results of a vanilla MLP network (i.e., the projection layer removed), where (c)(d) corresponds to the prediction of the same vanilla MLP with our stability projection layer. Since they are trained on the same dataset, the comparison illustrates the fact that without a stability projection layer, the prediction can grow unbounded leading to unreliable statistics.
> >
> > **Questions:**
> > 1. *Selection and generalization of Lyapunov functions: Although Lyapunov functions are very useful tools in control theory, it may not be easy to design and select appropriate Lyapunov functions in complex chaotic systems, especially in multidimensional or nonlinear systems. Many times, the form of the Lyapunov function needs to be manually designed or assumed based on the specific system, and may lack generalization. Is this approach applicable to a wider range of chaotic systems? In complex practical applications, finding a suitable Lyapunov function may require a lot of expert knowledge and manual design, which limits scalability. If the choice of Lyapunov function is not robust enough, the model may not be applicable to different types of chaotic systems.*
> >
> > - We thank the reviewer for this comment. We would like to clarify that the effect of the Lyapunov function is only to guide the trajectory to asymptotically converge to the level set that contains the attractor. It is true that chaotic systems observe complex behaviors, but these behaviors (as discussed thoroughly in Section 2) almost always occur on the strange attractor but not elsewhere for dissipative systems. Since the Lyapunov function has no effect after the trajectory enters the level set, and since the attractor is inside the level set, the ReLU function that incorporates the Lyapunov function information will be inactive thus has little or no influence on learning complex chaotic behaviors.
> > - Currently in numerical experiments, the choice of using quadratic functions is a simplified way to characterize the level set. It's worth noting that Proposition 1 and 2, where we derive dissipative conditions, do not specify the Lyapunov functions to be quadratic. Therefore, the framework could accommodate more complicated Lyapunov function designs. As mentioned above, since the goal is to guide the trajectory to converge inside the attractor and then rely on neural network to learn a model that reproduces invariant statistics, the choice of Lyapunov function is not closely connected to the capability of learning complex dynamics behaviors.
> > - Despite other important differences, the MNO paper can be effectively thought of as using a quadratic Lyapunov function with identity matrix in our framework. Their numerical examples show that a pre-defined unit ball can successfully encourage dissipativity while learning a variety of chaotic systems, therefore, we would expect a learned ellipsoid will not be restrictive to the learning problem.
> >
> > 2. *Design complexity and effectiveness of the projection layer: The projection layer is designed to ensure that the learned dynamic simulator is dissipative, which means that some parts of the neural network are explicitly designed to obey specific constraints (such as trajectory boundedness). However, the design and implementation of the projection layer can be complex ...... Excessive projection may limit the expressive power of the model, causing it to fail to capture some more complex dynamic behaviors.*
> >
> > - We thank the reviewer for this comment. Similar to comment to the previous question, the projection layer will have little influence since the ReLU activation in (5) will not be active inside the sublevel set. As discussed in Section 2, since the chaotic behaviors almost always happen on the attractor which is inside the sublevel set, the projection layer will not significantly impact the model's ability to capture complex dynamics behaviors.

---

> > > ### Author Response · Authors · 2024-11-16
> > > **Continued Responses**
> > >
> > > Question 3. *The paper claims to parameterize the Lyapunov function V as a quadratic function for simplicity and points out that the framework can accommodate any function that satisfies the requirements of Theorem 1. However, this statement is reasonable in theory but may have limitations in practical applications.*
> > >     1. *In practical applications, non-quadratic Lyapunov functions often significantly increase computational complexity. For example, parameterized as a neural network or a high-order polynomial will bring additional difficulties in gradient calculation and projection layer implementation.*
> > >     2. *Complex Lyapunov functions may be more difficult to maintain the required monotonicity and positive definiteness in the model, especially in high-dimensional chaotic systems.*
> > >     3. *Complex V forms may lead to numerical instability, especially in the training process, which is prone to gradient explosion or disappearance.*
> > > *Suggestions for improvement: If the authors hope to actually expand to a more general Lyapunov function form, the author can consider providing several experimental verifications of non-quadratic parameterized V to demonstrate the effectiveness of the framework in these cases. This work can further analyze the trade-offs between different V forms in terms of numerical stability and computational complexity to demonstrate the practical applicability and versatility of the framework.*
> > >
> > > - We thank the reviewer for the detailed comments and suggestions. Indeed, we plan to include more complicated parametrization choices for the Lyapunov function in our future work. However, as discussed in the previous two comments, the purpose of using the Lyapunov function sub-level set is to guide trajectory asymptotic convergence to a bounded region that contains the level set. Since the choice of Lyapunov function design does not affect learning inside the sub-level set, we will not expect using more complicated parameterization would improve the statistics prediction.
> > > - As alluded to earlier, the MNO paper can be effectively thought of as using a quadratic Lyapunov function with identity matrix in our framework. Their numerical examples show that a pre-defined unit ball can successfully encourage dissipativity while learning a variety of chaotic systems, therefore, we would expect a learned ellipsoid will not be restrictive to the learning problem.

---

> ### Comment · Reviewer_Y8Lv · 2024-11-20
>
> Thanks to the author's detailed response. Considering the method in the MNO paper, it has to be said that the novelty of this work is incremental and also lacks quantitative results to support their claims. Even so, I still look forward to seeing the improvements at the discussion stage. If all issues raised by the reviewers can be addressed, particularly high-dimensional equations that can exhibit turbulent characteristics, I can increase my score.

---

> > ### Author Response · Authors · 2024-12-01
> > **Response to Reviewer comments**
> >
> > We urge the reviewer to further investigate the substantial differences between our method and the MNO paper. The core novelty of our approach lies in its ability to guarantee dissipativity with respect to an explicitly constructed invariant set. In contrast, the MNO paper picks a pre-defined ball with no formal guarantee of trajectory convergence or boundedness to the chosen ball.
> >
> > More specifically, while the MNO paper pre-specifies the radius of the ball, our approach leverages control theory to formalize the notion of dissipativity. By constructing conditions that ensure dissipativity as an inductive bias within the architecture, our method allows the model to simultaneously learn the dynamics and the energy behavior with resepct to the invariant set. This represents a significant advancement in building theoretical guarantees for practical learning-based dissipative chaotic system modeling methods.
> >
> > For concerns with high-dimensional equations, we will show that our method can scale up reasonably in the next iteration of the paper. It is important to note that our approach does not deal with solution operators on a continuous spatial domain, as is the case in the setting of the MNO paper.
> >
> > While similar notions have been used in the MNO paper, our method provides strong formal guarantees and a fundamentally new perspective. The existence of the MNO paper does not diminish the contribution of our work. We hope the reviewer will recognize these differences and the broader impact of our contributions.

---

> > > ### Comment · Reviewer_Y8Lv · 2024-12-02
> > >
> > > I appreciate the author's effort to address my and other reviewers' concerns about the novelty. If our concerns about novelty are subjective, our concerns about the proposed method’s robustness are unanimous. Other reviewers also raised concerns about the effectiveness of this method for high-dimensional systems, and the authors agreed to add higher-dimension examples, e.g., 2D Kolmogorov flow on a 16-by-16 grid, to validate the scalability of their method. The fact that a month has passed and no revision, including the experiment, has been uploaded does make me less confident in the method proposed.

---

> > > > ### Author Response · Authors · 2024-12-03
> > > > **Responses**
> > > >
> > > > We would like to further clarify that the core novelty of our approach separates itself well from the literature by (1) enforcing dissipative conditions as hard constraints in the prediction model while other methods cannot (2) establishing a new control-theoretic perspective and deriving energy-based conditions that ensure a prediction model to be dissipative. These two key points are factual statements, rather than subjective claims.
> > > >
> > > > Additionally, consistent in all our responses, we proposed to add high-dimensional experiments in the camera-ready version. We would like to remind the reviewer Y8Lv that the time window for revision upload is two weeks rather than a month as the reviewer mentioned. We did not agree to finish the additional experiments in such a short time frame, as it was not reasonable. In this time frame, we have chosen to spend our effort in communicating our contributions to all reviewers and clarifying what additional experiments might be helpful to include in a camera-ready version.

---

### Official Review · Reviewer_P2tX · 2024-11-02

**Soundness:** 1
**Presentation:** 2
**Contribution:** 1
**Rating:** 3
**Confidence:** 5

**Summary:**

This paper introduces a Lyapunov-based approach to predict dissipative chaotic systems. In this paper, it learns a Lyapunov function enforcing the trajectories onto their invariant sets, subsequently generating trajectories within these invariant sets.

**Strengths:**

The paper addresses an intriguing problem in the realm of dissipative chaotic dynamical systems. The presentation is clear and straightforward, making the methodology easy to follow. The proposed theory effectively leverages Lyapunov functions to ensure the convergence of trajectories towards global strange attractors, establishing a meaningful connection between Lyapunov stability and chaotic system behavior.

**Weaknesses:**

### Theoretical Concerns
1. **Quadratic Lyapunov Function:**
   - The use of a quadratic form for the Lyapunov function is questionable. Chaotic systems exhibit highly complex behaviors (many chaotic systems are hyperbolic or partially hyperbolic [1]), and a simple quadratic Lyapunov function may not reliably guarantee global dissipativity—especially since the problem of constructing a global Lyapunov function for such systems remains unresolved. It is unclear how a constant positive definite matrix would suffice in this context.

[1] Hasselblatt, Boris, and Anatole Katok, eds. Handbook of dynamical systems. Elsevier, 2002.

2. **Optimization Challenges:**
   - Optimizing a quadratic Lyapunov function involves sum-of-squares optimization, which is related to the Hilbert 5th problem. This connection raises concerns about the scalability of the proposed algorithm, a point that the paper fails to adequately discuss. Furthermore, a quadratic Lyapunov function is inherently limited. For instance, certain chaotic systems are governed by PDEs with infinite-dimensional states, which would result in a positive definite matrix $Q$ of very high dimension, making optimization of such a matrix impractical.

3. **Invariant Set Representation:**
   - The paper claims that the sub-level sets of the Lyapunov function serve as global attractors. However, due to the quadratic nature of the Lyapunov function, these sub-level sets are high-dimensional ellipsoids, which may lack mathematical rigor. For example, the shape of many attractors may be irregular, such as the butterfly-shaped Lorenz-63, directly making the invariant sets as ellipsoids is too strong. The author proposed a joint learning way to train the model, but I question the training stability for chaotic systems, proving a loss curve figure is desirable.

4. **Difference with Existing Work:**
   - Research [1] presents a similar methodology by imposing a constraint to confine trajectories within an invariant set, treating the radius as a hyperparameter. The scale of this radius significantly impacts learning outcomes, yet the current paper does not address this aspect. Can you discuss the true difference between this work and [1], particularly in the novelty?

### Ergodicity Assumption
- **Lack of Guarantee:** After trajectories enter strange attractors, there is no theorem provided to guarantee ergodicity. The paper only reports the energy spectrum, raising concerns that trajectories may not fully explore the bounded attractor set, potentially conflicting with the ergodic assumption.
- **Missing Metrics:** A comparison metric, such as those based on transport distances like Maximum Mean Discrepancy (MMD) or Kullback-Leibler (KL) divergence, should be employed to measure the probability distribution on the invariant set. This is a common practice in learning algorithms to ensure invariant probability measures.

### Experimental Limitations
- **Loss Function:** The loss function in Line 368 is intended for learning local tangent vectors. However, for systems like Lorenz 63, the vector fields in the butterfly attractor exhibit extremely complex structures, such as fractal-like behaviors [2], which are also common in many other chaotic systems. In these scenarios, the vector fields can be highly intricate and even discontinuous, which I believe makes this problem NP-hard. I question the sample complexity and the feasibility of optimizing this loss function. Could you provide a plot of the training process and the loss value over time to illustrate its behavior?
- **Weak Experimental Section:** The experimental section is notably weak, lacking comparisons with other established algorithms such as those referenced in [1, 3]. It is better to compare comprehensively with the existing methods based on metrics such as short-term accuracy, long-term statistics (MMD or KL divergence).
- **Low-Dimensional Cases:** The experiments are confined to low-dimensional cases with minimal metrics. High-dimensional scenarios, including 2D cases like weather data or turbulent flows, are not explored, limiting the demonstration of the method's effectiveness.


[1] Li, Zongyi, et al. "Markov neural operators for learning chaotic systems." arXiv preprint arXiv:2106.06898 (2021): 2-3.

[2] Viswanath, Divakar. "The fractal property of the Lorenz attractor." Physica D: Nonlinear Phenomena 190.1-2 (2004): 115-128.

[3] Schiff, Yair, et al. "DySLIM: Dynamics Stable Learning by Invariant Measure for Chaotic Systems." arXiv preprint arXiv:2402.04467 (2024).

**Questions:**

1. **Redundancy in Theoretical Results:**
   - The main theorem and propositions presented in this paper are well-established in existing literature. The authors should directly cite relevant prior work instead of reiterating these foundational results without proper attribution.

2. **Ergodicity Guarantee:**
   - The paper does not provide a mechanism to ensure ergodicity, i.e., the ability of trajectories to fully explore the bounded invariant set based on its invariant measure once they reach the invariant set.

3. **Selection of Hyperparameters:**
   - The determination of the ellipsoid radius is inadequately addressed. The paper lacks a clear methodology for selecting an appropriate radius, raising concerns about the validity and reproducibility of the results.

4. **Training Stability and Loss Function:**
   - I question the training of vector fields in loss function, so could you provide a plot of the training process and the loss curves to illustrate its behavior?

---

> ### Author Response · Authors · 2024-11-15
> **We thank the reviewer for their comments and provide responses (Part I)**
>
> (the format is *original review comments* followed by our responses)
> **Weaknesses:**
> **Theoretical Concerns**
> 1. Quadratic Lyapunov Function:
> *The use of a quadratic form for the Lyapunov function is questionable. Chaotic systems exhibit highly complex behaviors (many chaotic systems are hyperbolic or partially hyperbolic [1]), and a simple quadratic Lyapunov function may not reliably guarantee global dissipativity—especially since the problem of constructing a global Lyapunov function for such systems remains unresolved. It is unclear how a constant positive definite matrix would suffice in this context.
> [1] Hasselblatt, Boris, and Anatole Katok, eds. Handbook of dynamical systems. Elsevier, 2002.*
>
> - We thank the reviewer for this comment. We would like to clarify that the effect of the Lyapunov function is only to guide the trajectory to asymptotically converge to the level set that contains the attractor. It is true that chaotic systems observe complex behaviors, but these behaviors (as discussed thoroughly in Section 2) almost always occur on the strange attractor but not elsewhere for dissipative systems. Since the Lyapunov function has no effect after the trajectory enters the level set, and since the attractor is inside the level set, the ReLU function that incorporates the Lyapunov function information will be inactive thus has little or no influence on learning complex chaotic behaviors.
> - Additionally, the choice of using quadratic functions is a simplified way to characterize the level set. Proposition 1 and 2, where we derive dissipative conditions, do not specify the Lyapunov functions to be quadratic. We use quadratic functions for simplicity, but the framework could accommodate more complicated Lyapunov function designs. As mentioned above, since the goal is to guide the trajectory to converge inside the attractor and then rely on neural network to learn a model that reproduces invariant statistics, the choice of Lyapunov function is not closely connected to the capability of learning complex dynamics behaviors.
> - Despite other important differences, the work [1] can be effectively thought of as using a quadratic Lyapunov function with identity matrix in our framework. Their numerical examples show that a pre-defined unit ball can successfully encourage dissipativity while learning a variety of chaotic systems, therefore, we would expect a learned ellipsoid will not perform worse than that.
>
> 2. Optimization Challenges:
> *Optimizing a quadratic Lyapunov function involves sum-of-squares optimization, which is related to the Hilbert 5th problem. This connection raises concerns about the scalability of the proposed algorithm, a point that the paper fails to adequately discuss. Furthermore, a quadratic Lyapunov function is inherently limited. For instance, certain chaotic systems are governed by PDEs with infinite-dimensional states, which would result in a positive definite matrix Q of very high dimension, making optimization of such a matrix impractical.*
> - We thank the reviewer for this comment. We would like to clarify that we do not use SoS to optimize our Lyapunov function. Instead, the Lyapunov function parameterized by $Q$ and $c$ is included in the loss function both in the dynamics prediction $\hat{x}_k^{(i)}$ terms which involves $f^*(x)$ defined (5) hence contains $Q$ and $c$, and in the regularization term. Therefore, we can use standard SGD to optimize $Q$ and $c$ via backpropagation.
> - As suggested by other reviewers, we are currently working on additional experiments that will be added to the camera-ready version. From the perspective of training neural networks with SGD, we would not expect computation issues as in a general learning context, large networks with more complicated parameterization structures have shown success in achieving the optimizaion goal reasonably well.
>
> 3. Invariant Set Representation:
> *The paper claims that the sub-level sets of the Lyapunov function serve as global attractors. However, due to the quadratic nature of the Lyapunov function, these sub-level sets are high-dimensional ellipsoids, which may lack mathematical rigor. For example, the shape of many attractors may be irregular, such as the butterfly-shaped Lorenz-63, directly making the invariant sets as ellipsoids is too strong. The author proposed a joint learning way to train the model, but I question the training stability for chaotic systems, proving a loss curve figure is desirable.*
> - We thank the reviewer for this comment. We would like to clarify that the learned sub-level sets are not claimed to serve as global attractors. Instead we use these sub-level sets as an outer-estimate of the attractor. The main purpose for sub-level sets of the Lyapunov function is to provide trajectory boundedness, learning an attractor outer-estimate is a byproduct.
> - As an example, in Fig.7-9, it is observed that the learned ellipsoids are reasonably tight outer-estimates for the attractor in practice.

---

> ### Author Response · Authors · 2024-11-15
> **Continued Responses**
>
> **Theoretical Concerns (cont'd)**
>
> 4. Difference with Existing Work:
> *Research [1] presents a similar methodology by imposing a constraint to confine trajectories within an invariant set, treating the radius as a hyperparameter. The scale of this radius significantly impacts learning outcomes, yet the current paper does not address this aspect. Can you discuss the true difference between this work and [1], particularly in the novelty?*
> - We would like to point out that despite the similar notions used in our paper and [1], our paper provides a different perspective that leverages control theory to learn the dynamics and the bounded region simultaneously, rather than using a predefined region regardless of its shape. Importantly, in [1] the bounded region is not guaranteed to be an invariant set, where as in our paper, we derive control theory inspired conditions that guarantee trajectory boundedness, which is crucial for obtaining meaningful statistics from the predicted trajectories.
> - More specifically, the distinctions between [1] and our approach can be summarized as follows:
>     1. The approach in [1] relies on prior expert knowledge to construct a ball that encourages dissipativity. The radius of the ball is a predefined hyperparameter and must be larger than the attractor, which is typically not known a priori. Instead, our method automates the choice of the level set, which eliminates the need of knowing attractor size before training.
>     2. To the best of our understanding, there is no formal guarantee for predicted trajectories to be bounded by the ball stated in [1]. In addition, the dissipativity condition is not stated formally in [1], which leaves the question of how to embed dissipativity into the learned model an open problem. In our paper, we provide a rigorous definition for dissipativity and enforce the condition with derived theoretical guarantees, thus bridged this gap left by [1].
>     3. Unlike [1] which adds the post-processing layer ***only in testing after the model is trained***, our stability projection layer is ***active during both training and testing***. Therefore, we avoid the requirement in [1] to preprocess the dataset into transient and ergodic phase, which allows our approach to directly work with any trajectory dataset without prior information about the attractor.
>     4. We would like to clarify that the goal of our paper is to provide a new perspective on understanding dissipativity as an energy decrease behavior, and proposes a network architecture that inherently builds the dissipative condition into the predictor structure. It addresses two main challenges: (i) formulating computable and back-propagation-ready dissipative conditions using Lyapunov functions (ii) enforcing the dissipative conditions explicitly through a closed-form projection. These challenges are key to provide formal trajectory boundedness guarantees, which are not addressed in [1].
>
>
> **Ergodicity Assumption**
> 1. *Lack of Guarantee: After trajectories enter strange attractors, there is no theorem provided to guarantee ergodicity. The paper only reports the energy spectrum, raising concerns that trajectories may not fully explore the bounded attractor set, potentially conflicting with the ergodic assumption.*
> - We thank the reviewer for this comment. To the best of our knowledge, there is no established theory that proves ergodicity in chaotic systems. The goal of our paper is to address trajectory boundedness rather than improving empirical statistical prediction.
> - To further address this concern, as the reviewer suggests, we will add more statistical evaluation metrics to validate invariant statistics preservation in the predicted trajectories.
>
> 2. *Missing Metrics: A comparison metric, such as those based on transport distances like Maximum Mean Discrepancy (MMD) or Kullback-Leibler (KL) divergence, should be employed to measure the probability distribution on the invariant set. This is a common practice in learning algorithms to ensure invariant probability measures.*
> - We are currently working on adding additional metrics to address this concern.

---

> ### Author Response · Authors · 2024-11-15
> **Continued Responses**
>
> **Experimental Limitations**
> 1. *Loss Function: The loss function in Line 368 is intended for learning local tangent vectors. However, for systems like Lorenz 63, the vector fields in the butterfly attractor exhibit extremely complex structures, such as fractal-like behaviors [2], which are also common in many other chaotic systems. In these scenarios, the vector fields can be highly intricate and even discontinuous, which I believe makes this problem NP-hard. I question the sample complexity and the feasibility of optimizing this loss function. Could you provide a plot of the training process and the loss value over time to illustrate its behavior?*
>
> - We thank the reviewer for this comment. To the best of our knowledge, there is no result in the literature that establishes the complexity of learning algorithms for general chaotic system modeling problems. We would like to respectfully remind the reviewer, as stated at the beginning of Section 2, the learning setting is to approximate a continuously differentiable function f: R^n\to R^n, where classical universal approximation problems suggest this class of functions can be approximated with an MLP to arbitrary accuracy.
> - Additionally, as the reviewer suggests, **we include the training loss history over epochs in the appendix**, which did not observed any convergence issue. **Please see Appendix B.4 for details.**
>
> 2. *Weak Experimental Section: The experimental section is notably weak, lacking comparisons with other established algorithms such as those referenced in [1, 3]. It is better to compare comprehensively with the existing methods based on metrics such as short-term accuracy, long-term statistics (MMD or KL divergence).*
>
> - As the reviewer suggests, we are working on providing additional statistics evaluations.
> - With regards to benchmarks, we make the following comments:
> (1)  	The goal of this paper is to propose a new method that provides formal trajectory boundedness guarantees with a learned energy function representation. The key innovations are the derived algebraic conditions for dissipativity and the stability projection layer in the network.  Comparing a model with the stability projection layer on (our model) and off (vanilla MLP) is sufficient to validate our approach. More specfically, comparing with reservoir computing methods such as the paper mentioned here is unfair, as these methods use a recurrent network structure that uses more history state information in its input.
> (2)  	We don’t make any claim on improving statistical evaluation than other methods. The benefit of having theoretical trajectory boundedness guarantee is to eliminate any possible trajectory rollout finite-time blowup, which is known to happen for most models in the current literature. Therefore, the experiments are designed to illustrate this point.
> (3)  	We choose the MLP as the “backbone” model due to its expressiveness in approximating a continuous function, which is all there is to learn for $f(x)$. Additionally, under our setting, it is equivalent to neural operator which has been shown to be successful in recent literature. Although not addressed in the scope of this paper, the stability projection layer could extend to more performant backbone models.
>
>
> 3. *Low-Dimensional Cases: The experiments are confined to low-dimensional cases with minimal metrics. High-dimensional scenarios, including 2D cases like weather data or turbulent flows, are not explored, limiting the demonstration of the method's effectiveness.*
> - We thank the reviewer for this comment, and we will provide additional experiments that validate our approach on high-dimensional chaotic ODEs in the camera-ready version.
>
> *[1] Li, Zongyi, et al. "Markov neural operators for learning chaotic systems." arXiv preprint arXiv:2106.06898 (2021): 2-3.
>
> [2] Viswanath, Divakar. "The fractal property of the Lorenz attractor." Physica D: Nonlinear Phenomena 190.1-2 (2004): 115-128.
>
> [3] Schiff, Yair, et al. "DySLIM: Dynamics Stable Learning by Invariant Measure for Chaotic Systems." arXiv preprint arXiv:2402.04467 (2024).*

---

> ### Author Response · Authors · 2024-11-15
> **Continued Responses**
>
> **Questions:**
> 1. *Redundancy in Theoretical Results:
> The main theorem and propositions presented in this paper are well-established in existing literature. The authors should directly cite relevant prior work instead of reiterating these foundational results without proper attribution.*
> - We thank the reviewer for this comment. Unlike conventional control literature that addresses asymptotic convergence to an equilibrium point, here the stability results are with respect to a sub-level set. Additionally, proposition 1 and 2 serve as building blocks for the main theorem that establishes theoretical trajectory boundedness which is a key contribution for our approach.
>
> 2. *Ergodicity Guarantee:
> The paper does not provide a mechanism to ensure ergodicity, i.e., the ability of trajectories to fully explore the bounded invariant set based on its invariant measure once they reach the invariant set.*
> - We thank the reviewer for this comment. As mentioned previously, to the best of our knowledge, there is no established theory that proves ergodicity in chaotic systems. Without an established theory for known chaotic systems, there is no feasible way to ensure ergodicity in any learned system model.
>
> 3. *Selection of Hyperparameters:
> The determination of the ellipsoid radius is inadequately addressed. The paper lacks a clear methodology for selecting an appropriate radius, raising concerns about the validity and reproducibility of the results.*
> - We thank the reviewer for this comment. As discussed in Section 4, the sub-level sets are characterized by learnable parameters Q and c, therefore, the shape of ellipsoid is learned simultaneously during training.
>
> 4. *Training Stability and Loss Function:
> I question the training of vector fields in loss function, so could you provide a plot of the training process and the loss curves to illustrate its behavior?*
>
> - We thank the reviewer for this comment. As the reviewer suggests, we have included the training loss history over epochs, which did not observed any convergence issue. Please see Appendix B.4 for details.

---

> ### Comment · Reviewer_P2tX · 2024-11-24
> **Response**
>
> Thank you for the authors' prompt response. However, some of your claims are incorrect, and the experiment lacks sufficient convincing evidence.
>
> 1. About constant matrix in Lyapunov function and high-dimensional cases.
>
> The framework appears limited when applied to infinite-dimensional PDEs, primarily due to constraints imposed by the matrix $Q$. For example, even for discretized Kolmogorov flow, experiments typically require resolutions of $64 * 64$ or even $256 * 256$ [1, 2, 3]. Optimizing a positive-definite matrix $Q = L^T L$ seems extremely hard for a $65536 * 65536$ matrix by a neural network in a $256 * 256$ Kolmogorov flow.
>
> I noticed that the authors are preparing experiments on a NS equation with $16 * 16$ grid, but such low-resolution cases provide limited information on the energy spectrum and statistical properties. A similar study on dissipative chaos was presented at ICLR 2025 [4], and if the authors could achieve such comprehensive experimental results with, as suggested in [4], I will consider increasing my score to 8.
>
> 2. About the ergodicity on chaotic system.
>
> The claim that no research proves the ergodicity of chaotic systems is inaccurate. Researchers aim to understand the long-term behavior of chaotic systems precisely to discover invariant measures on the phase space. For instance, the 2D Navier-Stokes (NS) equation has been proven ergodic [5, 6]. Similarly, systems such as the Lorenz and Kuramoto-Sivashinsky (KS) equations have also been shown to exhibit ergodic behavior empircally.
>
> [1] Schiff, Yair, et al. "DySLIM: Dynamics Stable Learning by Invariant Measure for Chaotic Systems." arXiv preprint arXiv:2402.04467 (2024).
>
> [2] Li, Zongyi, et al. "Markov neural operators for learning chaotic systems." arXiv preprint arXiv:2106.06898 (2021): 2-3.
>
> [3] Li, Zijie, Dule Shu, and Amir Barati Farimani. "Scalable transformer for pde surrogate modeling." Advances in Neural Information Processing Systems 36 (2024).
>
> [4] https://openreview.net/forum?id=Llh6CinTiy
>
> [5] Kupiainen, Antti. "Ergodicity of two dimensional turbulence." arXiv preprint arXiv:1005.0587 (2010).
>
> [6] Hairer, Martin, and Jonathan C. Mattingly. "Ergodicity of the 2D Navier-Stokes equations with degenerate stochastic forcing." Annals of Mathematics (2006): 993-1032.

---

> > ### Author Response · Authors · 2024-12-01
> > **Responses to Reviewer's comments**
> >
> > Regarding the problem formulation and scalability:
> >
> > 1. The goal of this paper is to propose a predictive model for finite-dimensional chaotic systems that is inherently dissipative. Despite the practical importance of PDEs like Kuramoto-Sivashinsky or Kolmogorov flow, we do not claim that our framework directly addresses learning PDE solution operators like the problem settings considered in [1, 2, 3]. The requirement that the framework should be scalable to produce dissipative ODE models that are of very high dimensions effectively approximating PDEs may not be suitable for the ODE learning setting under our consideration.
> >
> > 2. Regarding the scalability issues related to the size of $Q$, for experiment results similar to [4], the Kolmogorov flow example is constructed on a 64 by 64 grid, we will show that our architecture is able to scale up to the same dimension in the next iteration of the paper. Additionally, to ease computation burden, there are multiple ways to increase sparsity in the matrix $Q$, in the most simple case, using a diagonal matrix and only learning a vector of the same size as $x$.
> >
> >     More importantly, the quadratic function $V(x) = (x-x_0)^T Q (x-x_0)$ with matrix $Q$ is merely one way to parameterize the Lyapunov function. The framework is general enough to include other paramterizations to reduce the computation cost, not necessarily having to deal with the specific $n^2$ parameters in a full matrix $Q$. We will further investigate other parameterization methods to support this claim in the next iteration of our paper.
> >
> > Regarding the ergodicity assumption: We thank the reviewer for their helpful comments and providing pointers.
> >
> > We would also like to point out that despite existence of theoretical results, to our knowledge, most learning-based modeling approaches in the literature has no guarantees on ergodicity of the predicted trajectories. We acknowledge the importance of more statistical evaluation results in validating our method, but these results should all be in an empirical nature.

---

### Official Review · Reviewer_R8oE · 2024-11-02

**Soundness:** 2
**Presentation:** 3
**Contribution:** 2
**Rating:** 3
**Confidence:** 3

**Summary:**

This paper proposes a neural network architecture designed to learn dissipative chaotic dynamical systems with a guarantee of generating bounded trajectories. By integrating the control theory principles to enforce algebraic conditions ensuring that the learned dynamics converge to invariant measure. While the problem addressed is interesting and the theoretical guarantees looks reasonable, there are several concerns regarding the clarity of the method, the experimental evaluation.

**Strengths:**

The paper addresses the challenging problem of learning dissipative chaotic systems, which has broad implications in fields like weather prediction and fluid dynamics.The authors provide a theoretical basis for their approach, leveraging Lyapunov principle in control theory to ensure dissipativity and boundedness of the learned trajectories.

The experimental results, achieved through the rollout of predictions, converge to the invariant statistical sets of various chaotic systems, including the Lorenz63, Lorenz96, and the truncated Kuramoto–Sivashinsky equations.

**Weaknesses:**

The explanation of the proposed methodology, especially the construction of the stability projection layer and its practical implementation, lacks clarity.

Loss function:
It is stated that c is learned via the loss function, however, if I understand correctly, c is also exist in the loss function set up. if
c were to be preset rather than learned, it would constrain the size of M(c) from the beginning, potentially leading to suboptimal results if this predefined boundary doesn’t align well with the attractor in the data.

In this paper, "Markov neural operators for learning chaotic systems,  Advances in Neural Information Processing Systems, 2022"
They defined a ball to bound the trajectory, to me, it seems that you set up a Ellipsoid boundary other than the ball. Thus, the contribution is questionable.


Experimental:
- Despite important problem, this work only use low dimensional data sets to test the idea which is not convincing enough, several works can achieve this without
- Training data sets is very limited, with only 4-10 trajectories it is hard to believe that how effective the the method for the unseen data.
- Figure is misleading, Fig 4,5 a and b should be just MSE loss, and the author still name it as fstar which leads to confusion.
- The baseline used here is questionable, as it only compares against the Mean Squared Error (MSE) loss without incorporating the Lyapunov component. For instance, the paper "Stability Analysis of Chaotic Systems from Data" also successfully reconstructs the invariant measure. Without comparing the proposed method to other established approaches, it remains unclear how effective it truly is. Demonstrating its performance against a broader set of baselines would strengthen the argument for its effectiveness.

**Questions:**

1. please clarify if the c is learned or pre-set. if learned please describe how does it iteratively incorporated with the loss function.

2. Why the dimension of provided examples are very low? Does the model fits the high dimensional systems? at least 2D  Kolmogorov flow should be presented.

3. Does the training set and test sets are the same?

If authors can compare their method with the state-of-the-art model like Markov neural operators, it would be more convincing.

---

> ### Author Response · Authors · 2024-11-15
> **We thank the reviewer for their comments and provide responses to the comments (part I)**
>
> (the format is *reviewer comments* followed by our response)
> **Weaknesses**
>
> 1. *Loss function: It is stated that $c$ is learned via the loss function, however, if I understand correctly, $c$ is also exist in the loss function set up. if $c$ were to be preset rather than learned, it would constrain the size of $M(c)$ from the beginning, potentially leading to suboptimal results if this predefined boundary doesn’t align well with the attractor in the data.*
>
> - As stated in the paragraph under Figure 2, $c$ is one of the three learnable components. The loss function is a function of the learnable parameter $c$ because (1) $Vol(M(c))$ depends on c directly (2) the predicted trajectory samples $\hat{x}_k^{(i)}$ depends on c because the underlying dynamics $f^*(x)$, defined in (5), depends on $c$ when the ReLU term is active. Since the loss function explicityly depends on $c$, we can optimize/learn $c$ with stochastic gradient descent (SGD) through standard backpropagation.
>
> 2. *In this paper, "Markov neural operators for learning chaotic systems, Advances in Neural Information Processing Systems, 2022" They defined a ball to bound the trajectory, to me, it seems that you set up a Ellipsoid boundary other than the ball. Thus, the contribution is questionable.*
>
> - (For ease of exposition, we refer the mentioned work as "the MNO paper") We would like to point out that despite the similar notions used in our paper and the MNO paper, our paper provides a different perspective that leverages control theory to learn the dynamics and the bounded region simultaneously, rather than using a predefined region regardless of its shape.
> - More specifically, the distinctions between the MNO paper and our approach can be summarized as follows:
>     1. The approach in the MNO paper relies on prior expert knowledge to construct a ball that encourages dissipativity. The radius of the ball is a predefined hyperparameter and must be larger than the attractor, which is typically not known a priori. Instead, our method automates the choice of the level set, which eliminates the need of knowing attractor size before training.
>      2. To the best of our understanding, there is no formal guarantee for predicted trajectories to be bounded by the ball stated in the MNO paper. In addition, the dissipativity condition is not stated formally in the MNO paper, which leaves the question of how to embed dissipativity into the learned model an open problem. In our paper, we provide a rigorous definition for dissipativity and enforce the condition with derived theoretical guarantees, thus bridged this gap left by the MNO paper.
>      3. Unlike the MNO paper which adds the post-processing layer ***only in testing after the model is trained***, our stability projection layer is ***active during both training and testing***. Therefore, we avoid the requirement in the MNO paper to preprocess the dataset into transient and ergodic phase, which allows our approach to directly work with any trajectory dataset without prior information about the attractor.
>      4. We would like to clarify that the goal of our paper is to provide a new perspective on understanding dissipativity as an energy decrease behavior, and proposes a network architecture that inherently builds the dissipative condition into the predictor structure. It addresses two main challenges: (i) formulating computable and back-propagation-ready dissipative conditions using Lyapunov functions (ii) enforcing the dissipative conditions explicitly through a closed-form projection. These challenges are key to provide formal trajectory boundedness guarantees, which are not addressed in the MNO paper.
>
> **Experimental**
> 1. *Despite important problem, this work only use low dimensional data sets to test the idea which is not convincing enough, several works can achieve this without*
> - We thank the reviewer for this comment, and we will provide additional experiments that validate our approach on high-dimensional chaotic ODEs in the camera-ready version.
>
> 2. *Training data sets is very limited, with only 4-10 trajectories it is hard to believe that how effective the the method for the unseen data.*
> - We thank the reviewer for this comment, and we would like to point out that the amount of training data is sufficient as the same amount of data is used in (Li et al. 2022) (Jiang et al. 2024). Additionally, we’d like to clarify that the testing setup is obtaining a trajectory rollout by iteratively applying the model to predict the next state. Given that the state space is continuous, the states encountered during testing is almost never seen in the training dataset, which further confirms the “generalization” capability.

---

> > ### Author Response · Authors · 2024-11-15
> > **We thank the reviewer for their comments and provide responses to the comments (continued)**
> >
> > **Experimental**
> > 3. *Figure is misleading, Fig 4,5 a and b should be just MSE loss, and the author still name it as fstar which leads to confusion.*
> > - We thank the reviewer for this comment, and we will modify the legend for the MLP prediction to avoid confusion. We would like to clarify the purpose of Fig. 4 and 5 (a) (b) is to serve as comparison to our approach in Fig. 4/5 \(c\) (d). The difference between are that (a)(b) corresponds to the prediction results of a vanilla MLP network without the stability projection layer, where \(c\)(d) corresponds to the prediction of the same vanilla MLP with our stability projection layer. Since they are trained on the same dataset, the comparison illustrates the fact that without a stability projection layer, the prediction can grow unbounded leading to unreliable statistics. As we explained in Section 2, small MSE loss on the training data is not the goal in chaotic dynamics learning problems. Therefore, we chose to present the predicted trajectory and its energy spectrum.
> >
> > 4. *The baseline used here is questionable, as it only compares against the Mean Squared Error (MSE) loss without incorporating the Lyapunov component. For instance, the paper "Stability Analysis of Chaotic Systems from Data" also successfully reconstructs the invariant measure. Without comparing the proposed method to other established approaches, it remains unclear how effective it truly is. Demonstrating its performance against a broader set of baselines would strengthen the argument for its effectiveness.*
> > - We thank the reviewer for this comment, we would like to first clarify that there is no MSE loss comparison in our paper. Instead, the comparison study between a base MLP with or without stability projection layer is presented in Figure 4 and 5 to illustrate having theoretical guarantees for trajectory boundedness is crucial for obtaining meaningful statistics.
> > - We would also like to clarify as discussed in the MNO paper, for the finite-dimensional ODE problems considered in our paper, MLP is equivalent to the popular neural operator in terms of model architecture. Therefore, using MLP as an unconstrained base dynamics emulator is expressive enough.
> > - With regards to benchmarks, we make the following comments:
> > (1)  	The goal of this paper is to propose a new method that provides formal trajectory boundedness guarantees with a learned energy function representation. The key innovations are the derived algebraic conditions for dissipativity and the stability projection layer in the network.  Comparing a model with the stability projection layer on (our model) and off (vanilla MLP) is sufficient to validate our approach. More specfically, comparing with reservoir computing methods such as the paper mentioned here is unfair, as these methods use a recurrent network structure that uses more history state information in its input.
> > (2)  	We don’t make any claim on improving statistical evaluation than other methods. The benefit of having theoretical trajectory boundedness guarantee is to eliminate any possible trajectory rollout finite-time blowup, which is known to happen for most models in the current literature. Therefore, the experiments are designed to illustrate this point.
> > (3)  	We choose the MLP as the “backbone” model due to its expressiveness in approximating a continuous function, which is all there is to learn for $f(x)$. Additionally, under our setting, it is equivalent to neural operator which has been shown to be successful in recent literature. Although not addressed in the scope of this paper, the stability projection layer could extend to more performant backbone models.

---

> > > ### Author Response · Authors · 2024-11-15
> > > **We thank the reviewer for their comments and provide responses to the comments (continued)**
> > >
> > > **Questions**
> > > 1. *Please clarify if the c is learned or pre-set. if learned please describe how does it iteratively incorporated with the loss function.*
> > > - As discussed in the response to Weaknesses-1, $c$ is a learned parameter. The loss function is an explicit function of $c$, both in the dynamics prediction terms $\hat{x}_k^{(i)}$ which uses $c$ through the forward simulation of learned dynamics $\dot{x} = f^*(x)$ (where $f^*(x)$ is defined in (5)) and in the regularization term.
> > > 2. *Why the dimension of provided examples are very low? Does the model fits the high dimensional systems? at least 2D Kolmogorov flow should be presented.*
> > > - As discussed in the response to Experimental-1, to address this concern, we are currently working on additional experiments that will be added to the camera-ready version.
> > > 3. *Does the training set and test sets are the same?*
> > > - No, they are not the same. The training set contains trajectory segments as described in Section 4.2. The testing setting is to iteratively use the learned model $f^*(x)$ to produce a very long trajectory. As described in Section 5.1, "we roll out the model for 50000 time steps using 4th order Runge-Kutta numerical integration with a sampling rate of h = 0.01 [sec]", the testing trajectory is initiated from a randomly sampled state and obtained from simulating $\dot{x} = f^*(x)$. Given the chaotic nature of the dynamics we are studying, the testing dataset will almost never contain any input state seen during training.
> > > 4. *If authors can compare their method with the state-of-the-art model like Markov neural operators, it would be more convincing.*
> > > - Please see the responses to Weaknesses-2 and Experimental-4.

---

> > > > ### Comment · Reviewer_R8oE · 2024-11-24
> > > >
> > > > I thank author's quick response, I am patiently waiting for the author's experimental results for further discussions.

---

### Official Review · Reviewer_KyBv · 2024-11-03

**Soundness:** 4
**Presentation:** 4
**Contribution:** 3
**Rating:** 8
**Confidence:** 4

**Summary:**

This paper develops a novel approach to a neural-network-based model of dynamical systems.  A Lyapunov functional is learned and applied to the dynamics that guarantees that there are no unbounded trajectories.  Hence, stability of the system is preserved, which mitigates an issue that plagues many other neural surrogates --- instability of rolled out trajectories.

**Strengths:**

The authors present a novel idea that addresses a longstanding issue with many neural-network based surrogate models of chaotic dynamical systems, which is that neural surrogates often tend to be unstable.  The use of modifying dynamics through a Lyapunov functional to guarantee stability of the learned system is original and a conceptually interesting approach.  The authors' presentation is clear, and there is sufficient mathematical justification to demonstrate their claim of guaranteeing stability.

The results shown for several systems illustrate that the proposed method works well in practice (at least for the systems shown) -- the method indeed ensures stability, and also preserves invariant statistics.

This work is significant and would be of interest to many other researchers.  It also presents a future research direction because it can be extended in various ways.

**Weaknesses:**

- The ODE systems that this new method is applied to are all rather simple.  The most complex is a 8-degree-of-freedom truncation of the Kuramoto Sivashinsky equation.  I wonder, why not use a more complex problem, like the KS equation truncated to many more degrees of freedom, i.e. N=64 or 128?  That would preserve the spatial properties of the PDE much better.  The main reason I suggest it is that it leaves it unclear how well this method scales to more complex, more challenging problems and attractor geometries.

- Not a significant point, but I think some of the introduction can be tightened up, leaving a more concise paper or more room to discuss other points.  It may not be necessary to include so much exposition on chaotic systems, Lyapunov exponents, strange attractors, etc.

**Questions:**

Is there any effect on training, either in training convergence or in computational expense, by adding the Stability Projection?

---

> ### Author Response · Authors · 2024-11-16
> **We greatly appreciate the reviewer's acknowledgement and provide responses to their comments**
>
> We greatly appreciate the reviewer's time and acknowledgement of our contribution. Thank you for your interests and suggestions. In what follows, we provide responses corresponding to each comment.
>
> (format: *original review comment* followed by response)
>
> **Weaknesses:**
> 1. *The ODE systems that this new method is applied to are all rather simple. The most complex is a 8-degree-of-freedom truncation of the Kuramoto Sivashinsky equation. I wonder, why not use a more complex problem, like the KS equation truncated to many more degrees of freedom, i.e. N=64 or 128? That would preserve the spatial properties of the PDE much better. The main reason I suggest it is that it leaves it unclear how well this method scales to more complex, more challenging problems and attractor geometries.*
>
> - Thank you for your suggestion. We agree that the current set of experiments might leave concerns about scalability not properly addressed. We are working on numerical experiments for higher-dimensional systems, e.g., KS and/or 2D Kolmogorov flow on a 64/128 point spatial grids. We will include these experiments in the camera-ready version to further validate our approach.
>
> 2. *Not a significant point, but I think some of the introduction can be tightened up, leaving a more concise paper or more room to discuss other points. It may not be necessary to include so much exposition on chaotic systems, Lyapunov exponents, strange attractors, etc.*
>
> - Thank you for your suggestion. Our purpose for providing a self-contained and comprehensive background section is to improve the readability of our paper to a broader audience in the machine learning community. We will consider perhaps moving some parts of section 2 to the appendix to make room for more detailed discussions on our methodology and more numerical experiment validation.
>
> **Questions:**
> *Is there any effect on training, either in training convergence or in computational expense, by adding the Stability Projection?*
>
> - Thank you for your question. During training, we did observe that the model with the stability projection layer requires longer time to train (~4 times longer than the vanilla MLP) due to its additional gradient computation $\partial V/\partial x$ and projection ReLU implementation. However, the additional computational burden is not a significant concern, hence we believe it is well worth to pay the computation price for a predictor with boundedness guarantee and an invariant level set identification.
> - We have also included the training loss history in Appendix B.4 in the latest revision. The figures show that there is no convergence issue observed across models in all three examples.

---

### Official Review · Reviewer_zTer · 2024-11-03

**Soundness:** 3
**Presentation:** 3
**Contribution:** 3
**Rating:** 6
**Confidence:** 3

**Summary:**

The authors outline a novel approach to learn chaotic dynamics that ensures dissipativity. By deriving algebraic conditions for the dissipativity and implementing via a stability projection the authors claim to guarantee that the model generates bounded trajectories. The implementation approach captures the long-term statistics of various dynamical systems studied here.

**Strengths:**

- The work outlines novelty in the derived energy based algebraic conditions for dissipativity.
- Incorporating a regularisation loss of the invariant level set volume aids the model in detecting an outer estimate of the attractor.
  - The paper contains a comprehensive introduction to the topics considered.

**Weaknesses:**

- All figures could be greatly improved with proper legends and labels.
- Further numerical experiments on the validity of the approach and comparisons to other baselines would greatly strengthen the overall paper.

**Questions:**

- Is it possible to derive an alternative regulariser to the loss function to drive tighter bounds on the level sets?
- Do the authors foresee any difficulties when scaling to higher dimensional systems?

---

> ### Author Response · Authors · 2024-11-16
> **We thank the reviewer for their comments and provide responses**
>
> We appreciate the reviewer's acknowledgement of our paper's contributions and provides the following responses to their comments and questions.
>
> The format is *original review comments* followed by our responses.
>
> **Weaknesses:**
> 1. *All figures could be greatly improved with proper legends and labels.*
>
> - Thank you for the suggestions. We will improve the clarity and readability of legends and lables in all figures in the camera-ready version.
>
> 2. *Further numerical experiments on the validity of the approach and comparisons to other baselines would greatly strengthen the overall paper.*
>
> - Thank you for the suggestions. We are currently working on adding experiments on higher dimensional systems and on providing more statistical evaluation metrics to further validate our method, we will include them in the camera-ready version.
> - With respect to baselines, we would like to first clarify that the current experimental setup compares two models with stability projection layer on (our model) and off (vanilla MLP) which are trained on the exact same dataset. Since the goal of this paper is to propose a method that ensures trajectory boundedness, we believe this experiment setup aligns well with validating the effectiveness of our proposed method aligns with the goal. Additionally, in numerical simultions, we indeed observe that methods that are prone to finite-time blow-up lead to unreliable statistics.
> - Since there is no incentive in the current approach that encourages statistics matching, we will not expect our method to outperform other methods that explicitly consider matching statistics. However, the majority of other baseline methods typically do not have boundedness guarantee, which is prone to generate unreliable trajectories as the vanilla MLP case.
> - We agree with the reviewer that comparison with other baselines can still strengthen the paper, and we will try to implement the comparison in the camera-ready version.
>
> **Questions:**
> 1. *Is it possible to derive an alternative regulariser to the loss function to drive tighter bounds on the level sets?*
> - Thank you for the suggestion, one idea is to incorporate more statistics matching regularization to further encourage preservation of the invariant statistics, similar to the ideas in (Jiang et al. 2024), which might help identify tighter level sets.
> - Currently, from Figure 7-9, our level sets seem to be reasonably tight outer-estimates for the attractors.
> 2. *Do the authors foresee any difficulties when scaling to higher dimensional systems?*
> - We don't foresee any difficulties as learning finite-dimensional systems using MLP is not intrinsically a very difficult task, and scaling the matrix $Q$ up should not be a concern since in general larger networks with more complicated architectures and more parameters have been successfully trained.
> - We will add higher diemnsional system examples in the camera-ready version to address this concern.

---

> ### Comment · Reviewer_zTer · 2024-11-26
>
> I would like to thank the author(s) for addressing my comments. I look forward to seeing the new high-dimensional results before reassessing my rating.

---

### Comment · Area_Chair_yY7r · 2024-11-25
**Reviewers' Response**

Dear Reviewers,

As the author-reviewer discussion period is approaching its end, I would strongly encourage you to read the authors' responses and acknowledge them, while also checking if your questions/concerns have been appropriately addressed.

This is a crucial step, as it ensures that both reviewers and authors are on the same page, and it also helps us to put your recommendation in perspective.

Thank you again for your time and expertise.

Best,

AC

---

### Note · Authors · 2025-01-21

**Comment:**

Dear SACs and ACs,

We would like to formally request the withdrawal of our paper, "Learning chaotic dynamics with embedded dissipativity". While some reviewers clearly acknowledged the novelty and contributions of our work, other reviewers seem to have misunderstood our contributions relative to existing literature and conflated our approach with fundamentally different methods.

Our core contributions can be summarized as follows:
- We proposed a neural network architecture for learning chaotic dynamics that guarantees to generate bounded trajectories, addressing the known issue that many learning-based models are prone to finite-time blowup.
- Our approach simultaneously learns the dynamics and an energy representation from trajectory data. The learned dynamics model preserves invariant statistics over long trajectory rollouts, and the energy representation provides an outer-estimate for the attractor.
- Unlike other methods that rely heavily on hyperparameter tuning and prior knowledge, our trajectory boundedness guarantees leverage a control-theoretic perspective, which describes energy evolution over time, naturally connecting with dissipative behaviors in chaotic dynamics.

We plan to refine our manuscript to more clearly emphasize the distinct contributions of our approach and submit it to another venue. We remain confident in the novelty and significance of our work. As Reviewer KyBv highlighted, our method “addresses a longstanding issue with many neural-network based surrogate models of chaotic dynamical systems, which is that neural surrogates often tend to be unstable”.

Even though Reviewer KyBv and Reviewer zTer clearly recognize the novelty and significance of our contributions, there seems to be a fundamental misunderstanding in some of the other reviewers’ assessment of our work. Specifically, some reviewers appear to believe that our method must be applicable to very high-dimensional ODE problems approximating PDEs. While this is a valid consideration for practical applications and an interesting future direction, it is not the focus of our work, which prioritizes providing theoretical boundedness guarantees — a crucial but orthogonal contribution. Furthermore, the novelty of our approach in the control-theoretic perspective and its rigorous theoretical guarantees were missed by some reviewers, mischaracterizing our contributions as incremental relative to fundamentally different existing approaches.

Unfortunately, it seems unlikely that this misunderstanding can be adequately resolved within the limited rebuttal time frame. Despite our efforts to communicate our contributions clearly and our attempts to provide additional numerical experiments, the additional higher-dimensional experimental work requested by the reviewers is not entirely fair or relevant to the scope of our method.

We sincerely appreciate the ACs’ and reviewers' time and effort in handling our submission.

Best regards,
Authors of Submission 12230

**Withdrawal Confirmation:**

I have read and agree with the venue's withdrawal policy on behalf of myself and my co-authors.